# Radiationless mechanism of UV deactivation by cuticle phenolics in plants

Ana González Moreno [1], Abel de Cózar [2,3✉], Pilar Prieto [4], Eva Domínguez [5] & Antonio Heredia [1✉]

Hydroxycinnamic acids present in plant cuticles, the interphase and the main protective barrier between the plant and the environment, exhibit singular photochemical properties that could allow them to act as a UV shield. Here, we employ transient absorption spectroscopy on isolated cuticles and leaf epidermises to study in situ the photodynamics of these molecules in the excited state. Based on quantum chemical calculations on *p*-coumaric acid, the main phenolic acid present in the cuticle, we propose a model in which cuticle phenolics display a photoprotective mechanism based in an ultrafast and non-radiative excited state deactivation combined with fluorescence emission. As such, the cuticle can be regarded as the first and foremost protective barrier against UV radiation. This photostable and photodynamic mechanism seems to be universal in land plants giving a special role and function to the presence of different aromatic domains in plant cuticles and epidermises.

[1] Instituto de Hortofruticultura Subtropical y Mediterránea La Mayora, Universidad de Málaga - Consejo Superior de Investigaciones Científicas Departamento de Biología Molecular y Bioquímica, Universidad de Málaga, E-29071 Málaga, Spain. [2] Departamento de Química Orgánica I / Kimika Organikoa I Saila, Facultad de Química / Kimika Fakultatea, Universidad del País Vasco / Euskal Herriko Unibertsitatea (UPV/EHU) and Donostia International Physics Center (DIPC), P. K, 1072, 20018 San Sebastián – Donostia, Spain. [3] Ikerbasque, Basque Foundation for Science, Plaza Euskadi 5, 48009 Bilbao, Spain. [4] Departamento de Inorgánica, Orgánica y Bioquímica. Facultad de Ciencias y Tecnología Química-IRICA, Universidad de Castilla-La Mancha, 13071 Ciudad Real, Spain. [5] Instituto de Hortofruticultura Subtropical y Mediterránea La Mayora, Universidad de Málaga - Consejo Superior de Investigaciones Científicas, Departamento de Mejora Genética y Biotecnología, Estación Experimental La Mayora, Algarrobo-Costa, E-29750 Málaga, Spain. ✉email: abel.decozar@ehu.es; heredia@uma.es

Protection from UV radiation damage, especially the more energetically UV-B that reaches the surface of the earth, was one of the challenges that plants had to overcome during the transition from aquatic to terrestrial environments[1,2]. Most of the UV-B emitted by the sun is absorbed by the ozone layer and only a minor fraction reaches the Earth's surface. However, UV-B can damage DNA, proteins and lipids, generate reactive oxygen species (ROS), and impair biological processes related to photosynthesis, growth and development[3,4]. Plants respond to UV radiation by accumulating UV-screening compounds, mainly phenolic acids and flavonoids, in the vacuoles of epidermal cells[4]. Both classes of compounds display complementary absorption spectra in the UV range, with flavonoids mainly absorbing in the UV-A region and hydroxycinnamic acids in the more energetic UV-B range[5]. Intracellular accumulation of phenolic acids and flavonoids after UV-B exposure has been studied extensively and considered the mechanism of UV-B photoprotection from damage to internal tissues[1,3,4,6,7]. However, under natural conditions, a clear correlation between UV-B exposure and intracellular phenolic accumulation has not been established or the differences were too low to be considered of significance[1,8,9]. Moreover, this model does not consider how epidermal cells themselves are protected from UV-B damage.

The plant cuticle that covers outer epidermal cells represents the true interphase between the plant and the environment. It is composed of a lipid polyester named cutin, cell wall polysaccharides, waxes and phenolics. Waxes can have a dual location, either present on the surface (epicuticular waxes) or inside the cuticle (intracuticular waxes)[10]. The phenolic compounds identified in the cuticle are hydroxycinnamic acids and, in some species, flavonoids[10]. These compounds have been identified in the epi and intracuticular wax fraction[11], trapped inside the cutin matrix, and covalently bonded to cutin monomers[10,12]. Nevertheless, despite its obvious importance as a barrier biopolymer with significant biophysical properties, from controlling water loss to conferring biomechanical support and acting as a thermal regulator[10], the optical properties of plant cuticles have received scarce attention. Surface reflectance, associated with epicuticular waxes or cuticle surface nanoridges, has been studied in relation to colour, gloss and pollinator attraction in a few species[13–15]. However, less attention has received cuticle phenolics, despite being located at the true interphase between the plant and the environment and their ability to attenuate UV transmission[16,17]. This phenolics compounds present in the cuticle have already been shown to play important biophysical roles conferring mechanical resistance and modifying water transport across the cuticle[10].

The phenolic acids identified in the cuticle of different species are p-coumaric, ferulic, caffeic and p-hydroxybenzoic acids, with p-coumaric acid present in all the species analyzed[12,18,19]. The first three compounds have in common the presence of a double bond in the acyl chain that can be in trans–cis conformation, thus allowing a possible double bond isomerization (shown in Fig. 1a). Photoisomerization of p-coumaric acid is behind the signal transduction of the bacterial yellow protein[20] and can provide a stable photoprotective mechanism against UV radiation. In vitro photochemical analyses of several cinnamic acids have shown their ability to perform an ultrafast isomerization process in a liquid state upon UV absorption[21–25]. This liquid environment could approximate a cytoplasm environment but it is very different from a solid matrix such as the cuticle. Here we employ for the first time ultrafast transient spectroscopy in solid biological samples to study the photodynamics of cuticle phenolics highlighting the crucial role of the cuticle in plant protection against UV radiation. Additionally, quantum chemical computational analyses, using p-coumaric acid, provide a mechanistic model of photoprotection based on a non-radiative excited-state deactivation of these phenolics within the cuticle.

## Results

**Optical behaviour of isolated cuticles.** Transmittance spectra of isolated cuticles of different species displayed a remarkable reduction in the UV-C (200-280 nm) and UV-B range (280–315 nm) (Fig. 1b). Notable differences were observed among species. A strong reduction in UV-B able to pass through the cuticle was observed in most species, between an average of 0.3% transmittance for *C. annuum* to 12% for *H. helix*. However, the two species with a very low amount of cuticle, *B. vulgaris* and *B. oleracea*, showed a much higher transmittance, reaching an average of 40 and 48% respectively, within the UV-B region. *I. germanica* showed an intermediate behaviour with 28% transmittance in UV-B. These results are similar to those reported for cuticles of woody species and some fruits[16,17]. This ability of the cuticle to drastically impair UV light transmission partially continued within the UV-A region (315–400 nm) until ~330 nm for some species. Within the UV-A range, transmittance increased exponentially until reaching the maximum in the visible light range (400–800 nm), varying between 60% for *A. americana* and 90% for *B. vulgaris*.

Two optical parameters, reflectance and absorbance, can be responsible for this notable UV protection of the cuticle. In general, the reflectance was low within the UV region, with values varying between 3–10% in UV-B depending on the species, regardless they were fruit or leaf cuticles (Fig. 1b). Around 300–400 nm reflectance increased to its maximum, 10–35% depending on the species, within visible light. Epicuticular wax disposition and crystallisation, as well as the presence and density of indument, are chiefly responsible for light reflection and could be behind the observed differences among species[13,26]. It should be noted the low reflectance observed in grapefruit cuticles despite the massive epicuticular wax accumulation present[27]. This could be related to triterpenoid acids being the main epicuticular wax components, whereas paraffins are minor components. Contrary to reflectance, cuticle absorbance was maximum at the UV range and explained the strongly attenuated, and in some instances almost negligible, transmittance. A broadband with two maxima around 220–240 nm and 280–310 nm was observed in the UV region of most cuticles (Fig. 1b). Interestingly, *B. oleracea*, *B. vulgaris* and *I. germanica* seemed to have a band at wavelength <200 nm with the band around 280–300 nm present as a shoulder. Around 350 nm absorbance strongly decayed and little to none was detected within the visible light range. This UV-B absorbance is assigned to the phenolic compounds present in the plant cuticle. Of the different phenolics present in plant cuticles, *p*-coumaric acid has been identified ubiquitously and in some species as the only cinnamic acid derivative present in the cuticle[28–31]. In this sense, absorption spectra of a *p*-coumaric acid solution displayed two major peaks in the UV region that agreed well with those observed in the cuticles (Supplementary Fig. 1). The amount of cuticle phenolics showed variability among species (Fig. 1c). Thus, the highest reduction in UV transmittance, and therefore high absorbance, was observed in the species with a higher amount of phenolics. At the same time, *B. oleracea* and *B. vulgaris* showed the lowest amount of phenolics, in agreement with their high transmittance. A non-linear regression fit to a power function was performed between cuticle UV-B absorbance and the amount of cuticle phenolics (Supplementary Fig. 2) showing a significant relationship between both variables.

Fluorescence emission is a well-known mechanism of energy dissipation in plants. Plant cuticles exhibited blue light autofluorescence after UV excitation (Supplementary Fig. 3). This

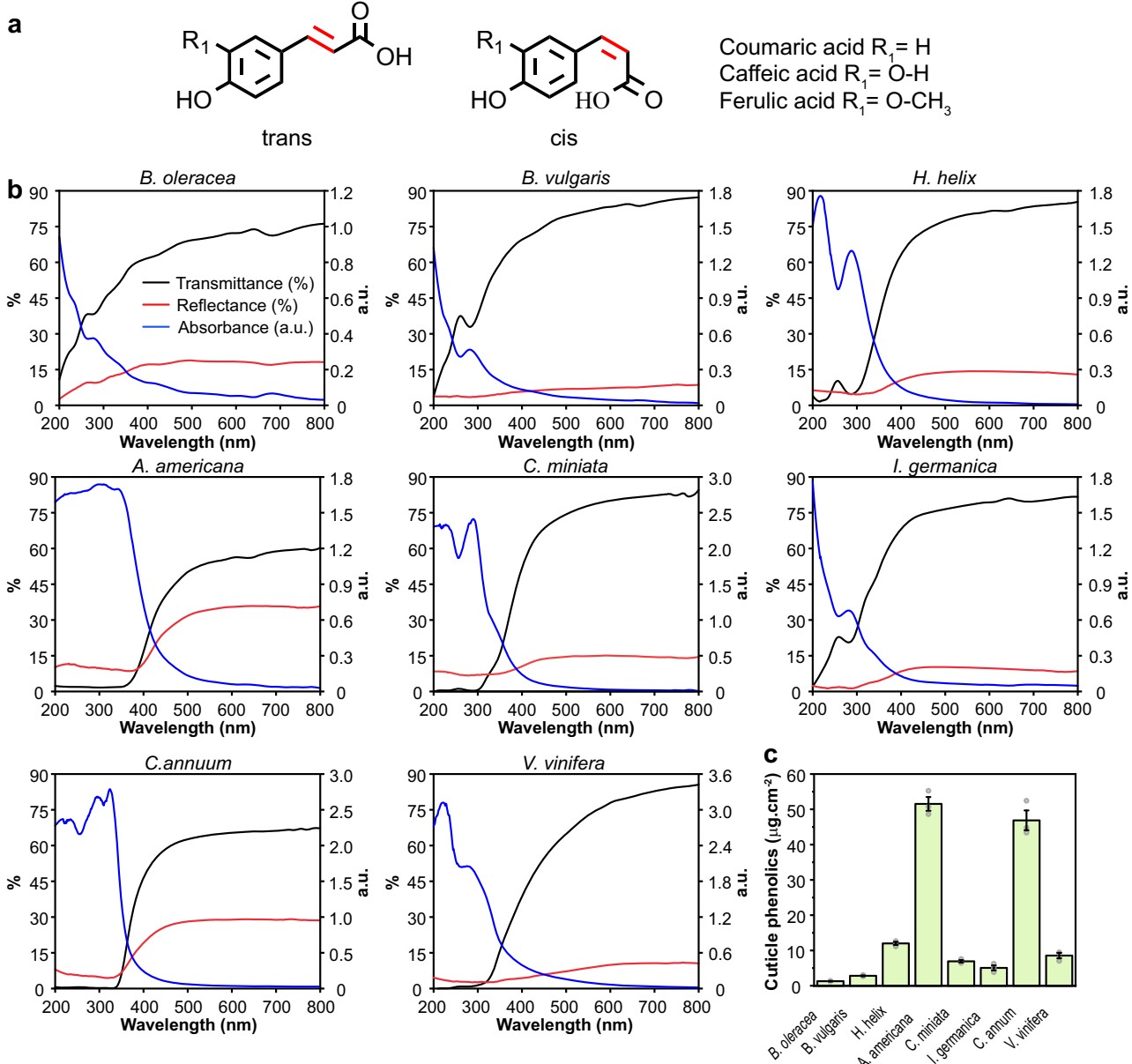

**Fig. 1 Cuticle phenolics and optical properties associated with the cuticle. a** *Trans* and *cis* chemical structure of cinnamic acids. In red the double bond is responsible for the *trans-cis* isomerization. To the right, substituents of the different cinnamic acids reported for cuticles of various species are indicated. **b** Transmittance, reflectance and absorbance spectra of isolated cuticles from different species. *Brassica oleracea*, *Beta vulgaris*, *Hedera helix*, *Agave americana*, *Clivia miniata* and *Iris germanica* leaf cuticles. *Capsicum annuum* and *Vitis vinifera* fruit cuticles. *n* = 4 biologically independent samples. **c** Amount of cuticle phenolics per surface unit. Data were represented as means ± s.e. *n* = 3 biologically independent samples.

weak fluorescence was mainly located to the outermost part of the cuticle and in some species, such as *C. annuum*, present in the whole cuticle. Fluorescence emission spectra of the isolated cuticles showed a low, broad and scattered band in the blue-cyan spectral region, in some instances extended to the green region, (Supplementary Fig. 4) and with fluorescence lifetimes around 2–8 ns, depending on the species. However, given the modicum of fluorescence observed in the cuticles and the known low fluorescence quantum yield of *p*-coumaric acid and other cinnamic acid derivatives[32], another mechanism of energy dissipation must be responsible for the deactivation of cuticle phenolics after UV absorption.

**Effect of the environment on *p*-coumaric acid isomerization.**
Transient absorption spectroscopy (TAS) maps of *p*-coumaric

acid in solution and in different solid environments are shown in Fig. 2a. The major advantage of this technique is the possibility to detect dynamic processes that occur below the nanosecond time scale (detection limit of traditional techniques) after excitation at a given wavelength[33]. A detailed explanation of the technique can be found in the Methods section and Supplementary Fig. 5. In a methanol solution, *p*-coumaric acid showed a positive and fast decaying band (Fig. 2a) between 340–450 nm that decayed after a few picoseconds and corresponded to an excited state absorption (ESA) process[34]. A stimulated emission (SE) was also observed. It has been reported that nano, micro and macromolecular domains have a significant influence on the stability and geometry of the excited state[22], and hence, *p*-coumaric acid was also analysed in two solid environments: powder and an alkylcoumarate oligomer where the cinnamic acid is esterified to a polyhydroxy alkyl chain.

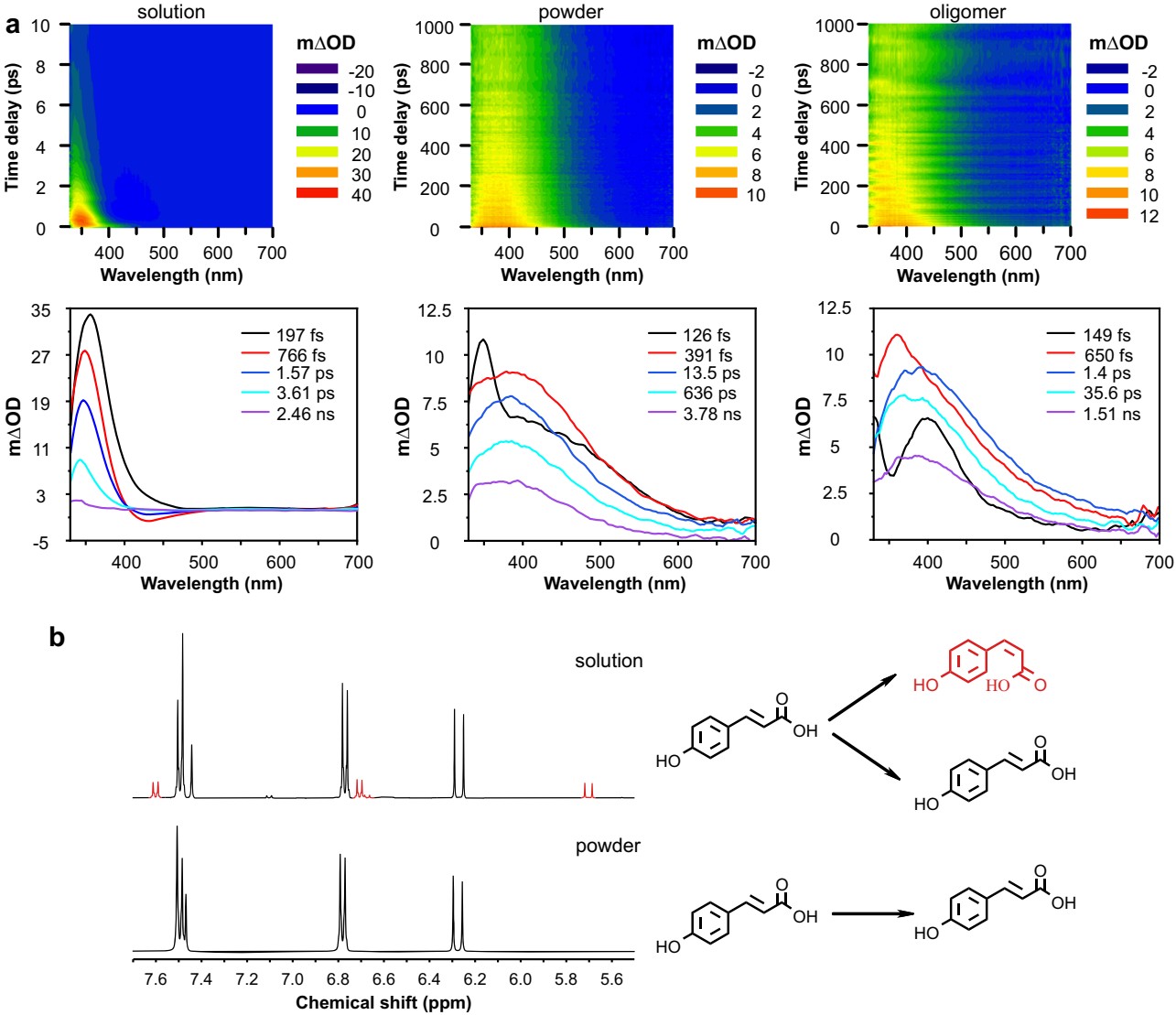

**Fig. 2 Transient absorption spectra (TAS) of *p*-coumaric acid in different environments. a** TAS of a methanol solution of *p*-coumaric acid ($10^{-3}$ M), powder *p*-coumaric acid and a polyalkylcoumarate oligomer. Top, heat maps displaying changes in optical density (mΔOD) with time and wavelength after a pump pulse of 300 nm. Bottom, TAS graphs display the evolution of differences in absorbance with wavelength at given delay times. **b** NMR spectra after 5 h of continuous irradiation at 300 nm of *p*-coumaric acid solution and solid samples. Red peaks in the solution sample are characteristic of the *cis*-isomer whereas those in black are of *trans*-isomer. To the right, a schematic representation of the processes occurring during sample irradiation. In black *trans p*-coumaric acid, in red *cis p*-coumaric acid.

These environments better resemble the two possible locations of cinnamic acids in the cuticle, trapped inside the matrix as free molecules or esterified to the cutin polymer or to very long chain alcohols. TAS reflection mode was employed to measure the spectra of solid samples (see Methods and Supplementary Fig. 5 for more details). Spectra recorded with this mode were noisier and broader due to light scattering and sample self-absorption. TAS comparison of liquid and solid samples rendered substantial differences. In both solid samples, powder *p*-coumaric acid and alkylcoumarate oligomer, one band was identified (Fig. 2a). This band, around 350–550 nm, was broader than the main band identified in liquid *p*-coumaric acid, and showed a much slower decay with time, reaching hundreds of picoseconds.

Kinetic decay analysis of the main TAS bands allowed the identification of three different processes occurring in the excited state and the calculation of their corresponding lifetimes (Table 1). These results agree with those reported in the literature for cinnamic acids[22,34] with the exception of an initial time, $\tau_1$,

within the femtosecond range, that could not be accurately resolved due to the instrument response function (IRF) value (see Methods). These different times have been assigned to conformational changes of cinnamic acid molecules during the *trans-cis* isomerization process in the excited state[23]. $\tau_1$ corresponds to an initial and very fast geometry relaxation of the molecule in the excited state $S_1$ and is associated with the Franck–Condon principle. $\tau_2$ and $\tau_3$ can be associated with the *trans-cis* isomerization process. In particular, $\tau_2$ relates to the vibrational cooling in the excited state and likely correspond to motion along the $\pi\pi^*$ state, while $\tau_3$ describes the internal conversion to the ground state along the photoisomerization coordinate[35]. Finally, a fourth time, $\tau_4$, present in some instances, has been ascribed to long-lived species with lifetimes of nanoseconds; in the case of the solution of *p*-coumaric acid, this long lifetime is ascribed to the presence of the cis-isomer. The estimated lifetimes for the *p*-coumaric acid solution are in accordance with those reported for other cinnamate systems[23,24] thus suggesting a very fast *trans-cis*

**Table 1 Sequential lifetimes of the excited state processes.**

| Sample | $\tau_2$ (ps) | $\tau_3$ (ps) | $\tau_4$ (ns) |
|---|---|---|---|
| *p*-coumaric acid solution | 0.25 ± 0.11 | 1.55 ± 0.11 | >ns |
| *p*-coumaric acid powder | 8.98 ± 0.20 | 367.11 ± 5.03 | >ns |
| *p*-coumaric acid oligomer | 42.07 ± 1.11 | 438.57 ± 18.21 | >ns |
| *Brassica oleracea* | 4.52 ± 0.13 | 249.33 ± 4.16 | >ns |
| *Beta vulgaris* | 46.55 ± 0.93 | 689.70 ± 16.36 | >ns |
| *Hedera helix* | 6.55 ± 0.29 | 260.00 ± 3.42 | >ns |
| *Agave americana* | 10.93 ± 0.22 | 284.65 ± 3.39 | >ns |
| *Clivia miniata* | 15.92 ± 0.78 | 365.47 ± 10.14 | >ns |
| *Iris germanica* | 17.26 ± 0.74 | 407.33 ± 12.18 | >ns |
| *Capsicum annuum* | 5.02 ± 0.17 | 249.31 ± 3.15 | >ns |
| *Vitis vinifera* | 9.61 ± 0.33 | 207.08 ± 3.06 | >ns |
| *Adiantum raddianum* | 7.48 ± 0.26 | 319.38 ± 3.60 | >ns |
| *Cycas revoluta* | 3.23 ± 0.12 | 97.13 ± 4.69 | >ns |
| *Araucaria bidwillii* | 6.31 ± 0.11 | 362.20 ± 6.44 | >ns |

Lifetimes (means ± errors associated with the calculated lifetimes) were calculated after sequential fitting analyses of the corresponding TAS of each sample. Fitting residuals and evolution-associated decay spectra (EADS) resulting from the TAS sequential fitting analyses are shown in Supplementary Figs. 12, 13 and 14. TAS transmittance mode was employed for *p*-coumaric acid solution and reflection mode for the rest of the samples. TAS was carried out on isolated cuticles of *B. oleracea*, *B. vulgaris*, *H. helix*, *A. americana*, *C. miniata*, *I. germanica*, *C. annuum* and *V. vinifera* and on leaf epidermises of *A. raddianum*, *C. revoluta* and *A. bidwillii*.

isomerization process (~2 ps) of *p*-coumaric acid in a liquid environment. The delayed decay observed in both solid environments in comparison with the fast deactivation observed in solution is reflected in the larger lifetimes estimated for both solid environments (Table 1). Thus, a notable increase in $\tau_2$ and $\tau_3$ times was observed, especially in the oligomer. This is probably due to the reduced degrees of freedom of the molecules in a solid matrix, thus reinforcing the strong dependence of the isomerization process on the molecular environment, as it has been reported for sinapoyl malate in high-viscosity solutions and polymer films[36,37].

To examine if a *trans-cis* isomerization occurred in liquid and solid samples, $^1$H-NMR spectra of a solution and powder *p*-coumaric acid after UV irradiation were recorded (Fig. 2b). After irradiation, the *p*-coumaric acid solution displayed resonances (Fig. 2b in black) characteristic of *trans*-isomer as well as additional ones (Fig. 2b in red) distinctive of the *cis*-isomer, in agreement with the reported literature[23,24]. COSY analysis was employed to confirm that these resonances corresponded to the *cis*-isomer. However, these characteristic resonances of the *cis*-isomer were not present in the irradiated *p*-coumaric acid powder sample. This is a clear indication that *trans-cis* isomerization can occur in solution, where molecules have a high degree of freedom, but not in a solid environment where the molecular movements are much more restricted.

**Plant cuticle photodynamics.** TAS spectra of isolated cuticles from various plant species are shown in Fig. 3. Despite the complex and composite nature of the plant cuticle matrix, the behaviour observed was very similar to that of *p*-coumaric acid solid samples. The main band (red and yellow in Fig. 3) was located between 340–450 nm in all cuticle samples with two exceptions, *B. vulgaris* where the region was broader, 340–500 nm, and *I. germanica* that showed a band-shift towards 400–550 nm. This main band slowly decayed with wavelength until 550–600 nm, depending on the species (green-blue transition in Fig. 3). Again, in *B. vulgaris* and *I. germanica* this decay was extended to higher wavelengths. The

lifetimes associated with this band varied among the different cuticle species (Table 1). In general, most of the species showed a $\tau_2$ similar or even lower than powder *p*-coumaric acid. The exceptions were *I. germanica* and *C. miniata* with values almost intermediate between the powder and the oligomer and *B. vulgaris* with a $\tau_2$ similar to that of the oligomer. The $\tau_3$ time followed a behaviour similar to that of $\tau_2$, with most species displaying values similar or lower than powder *p*-coumaric acid, and *B. vulgaris* considerably higher values. These results are an indication of phenolics present in an environment with a certain degree of freedom in most cuticles studied whereas in *B. vulgaris* these phenolics could be located in more restricted environments. It should be mentioned that *B. vulgaris* cuticle is mainly composed of cutan[38], a rigid and non-hydrolysable polymer matrix derived from unsaturated fatty acids, cross-linked with ether and peroxide bonds, with an aromatic domain[39]. This polymer rigidity would hinder molecular movements within the matrix and could explain the notably high $\tau_2$ and $\tau_3$ lifetimes detected in *B. vulgaris*, similar to those observed for the alkyl-coumarate oligomer.

TAS reflection mode was additionally tested in fresh leaf tissues of two gymnosperms and a fern to analyse cuticle photodynamics. The low laser penetration into the sample could be employed to study only processes that occur in or close to the surface without the need to isolate the cuticle. A band between 350–450 nm with an extended decay until 500–600 nm was observed (Supplementary Fig. 6), in agreement with the results obtained for isolated cuticles. In *C. revoluta* and *A. bidwillii* this band was slightly restricted to the 350–420 nm region. Lifetime calculations showed $\tau_2$ values similar to those obtained for isolated cuticles and within the range of powder *p*-coumaric acid or lower, as it was the case of *C. revoluta*. The $\tau_3$ times were equivalent to those obtained for isolated cuticles and powder *p*-coumaric acid, again with the exception of *C. revoluta* that displayed lower values. This indicates that in this species both photodynamic processes seem to occur faster. From the results obtained it is clear that, despite the differences in lifetimes and bandwidth observed among the different species, which could be the consequence of different phenolic acid fractions, their location within the cuticle, whether they are chemically bonded or trapped, and/or changes in the matrix itself, a common process is detected in all the studied species.

**Mechanism of phenolic UV deactivation.** As it was shown above, *trans-cis* isomerization of *p*-coumaric acid was not observed in a solid molecular environment. Hence, it would be extraordinary unlikely to happen in the cuticle, a more complex solid matrix where phenolics are trapped or covalently bonded, especially when lifetimes, close to those obtained for solid environments, are considered. Therefore, the mechanism behind their non-radiative deactivation and consequent plant photoprotection needs to be investigated. TD–DFT[40] and CASSCF[41] based theoretical analyses of the *p*-coumaric acid photochemistry were carried out. These studies allowed to explore the energy surfaces of the ground and excited states of *p*-coumaric acid (TD–DFT) and analyse the conical intersections (CASSCF). Additionally, dynamic electron correlation has been considered by means of CASPT2 single point energy calculations of TD–DFT or CASSCF optimised geometries[42]. Two environments were considered for the analyses: a solvated one to simulate liquid conditions and a minimalistic model of solid environment to mimic the cuticle scenario (for computational details see Methods). For clarity purposes, *trans* and *cis* isomers will be represented by a $\theta_{C1-C2-C3-C4}$ dihedral angle (between the phenyl ring and the carboxyl group) of 180° and 0°, respectively.

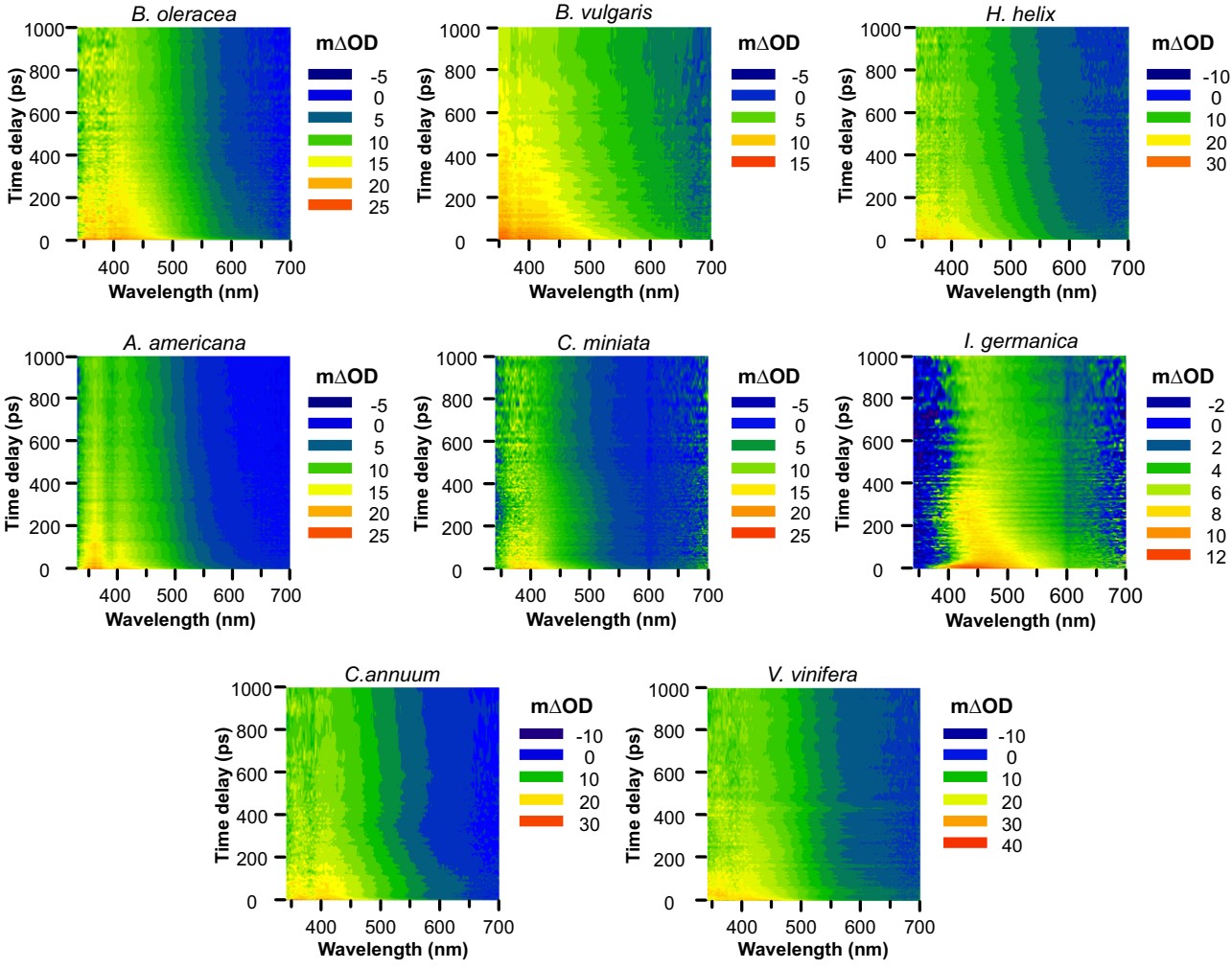

**Fig. 3 Transient absorption spectra (TAS) of isolated plant cuticles.** Heat maps of cuticles from different species displaying differences in optical density (m$\Delta$OD) with time and wavelength after a pump pulse photoexcitation of 300 nm. *Brassica oleracea, Beta vulgaris, Hedera helix, Agave americana, Clivia miniata* and *Iris germanica* leaf cuticles. *Capsicum annuum* and *Vitis vinifera* fruit cuticles.

For a thermal *trans-cis* isomerization process in the basal electronic state, starting with the *trans*-isomer (180°), the energy would rise as the dihedral angle decreases, reaching the geometry of the transition state (s-TS-$S_0$) at values close to 90°. At that point, the maximum energy point in the energy profile, the carboxyl group is perpendicular to the phenyl ring plane. Once the s-TS-$S_0$ is reached, the energy decreases as the dihedral angle goes from 90° to 0° and the *cis*-isomer is formed. Inspection of the *p*-coumaric acid frontier molecular orbitals showed that the ground state corresponds to a singlet state ($S_0$) with an electronic configuration $(\pi)^2(\pi^*)^0$ where frontier electrons are located in $\pi$-symmetry orbitals, which are characteristic of all unsaturated and aromatic organic compounds (Fig. 4a). UV photoexcitation will promote one frontier electron to a higher energy orbital, thus reaching the first singlet excited state $S_1$[43] with electronic configuration $(\pi)^1(\pi^*)^1$. This electron promotion is almost instantaneous (vertical excitation), leading to an excited electronic state with the geometry of the ground state (point b in Fig. 4b, c). If the lifetime of the excited state is long enough, the molecule will relax in the excited state (point c in Fig. 4b, c). This process is related to the first time, $\tau_1$, identified in the TAS analysis. Calculations including dynamic electron correlation showed a barrierless isomerization process in the excited state for the liquid (s) scenario, and a reduction in the energy activation and Gibbs free energy activation barriers (c.a. 9 kcal mol$^{-1}$) for the in solid

(sp) state (s-Ea-S1and sp-Ea-S1in Fig. 4b, c and Supplementary Tables 1 and 2)[44]. Remarkably, calculations indicate that dynamic electron correlation has a significant effect on the vertical excitations energy, and in the shape of the $S_1$ curve in solution but only a small effect in the general shape of the $S_0$ and $S_1$ curves in solid state[45]. On the other hand, exploration of the $S_0$ and $S_1$ energy surfaces revealed different shapes and allowed the identification of intersection points between both electronic states that is, conical intersections[46] (CI, Supplementary Table 3 and Supplementary Fig. 7), in both environments. After the initial relaxation in the excited state, the molecule continued changing its geometry to adopt a conformation close to that of the CI. This step could be associated with the $\tau_2$ reported in the TAS analysis. At this point, the electron returned to the ground electronic configuration ($S_0$) following a non-radiative pathway. This last step can be related to $\tau_3$ calculated from TAS.

CASSCF calculations in a solvated (s) environment displayed a CI $S_0$-$S_1$ geometry (s-CI, $\theta_{C1-C2-C3-C4} = 90.3°$) very close to that of s-TS-$S_0$ ($\theta_{C1-C2-C3-C4} = 94.5°$) (Fig. 4d left, see Supplementary Fig. 8 for further details on the transition structure geometries). Thus, the geometrical similarity between the conical intersection in a solvated environment (s-CI) and its transition state of the ground state (s-TS-$S_0$) would facilitate reaching the TS structure without additional energy required to overcome the isomerization barrier in the basal electronic state. Therefore, during the decay

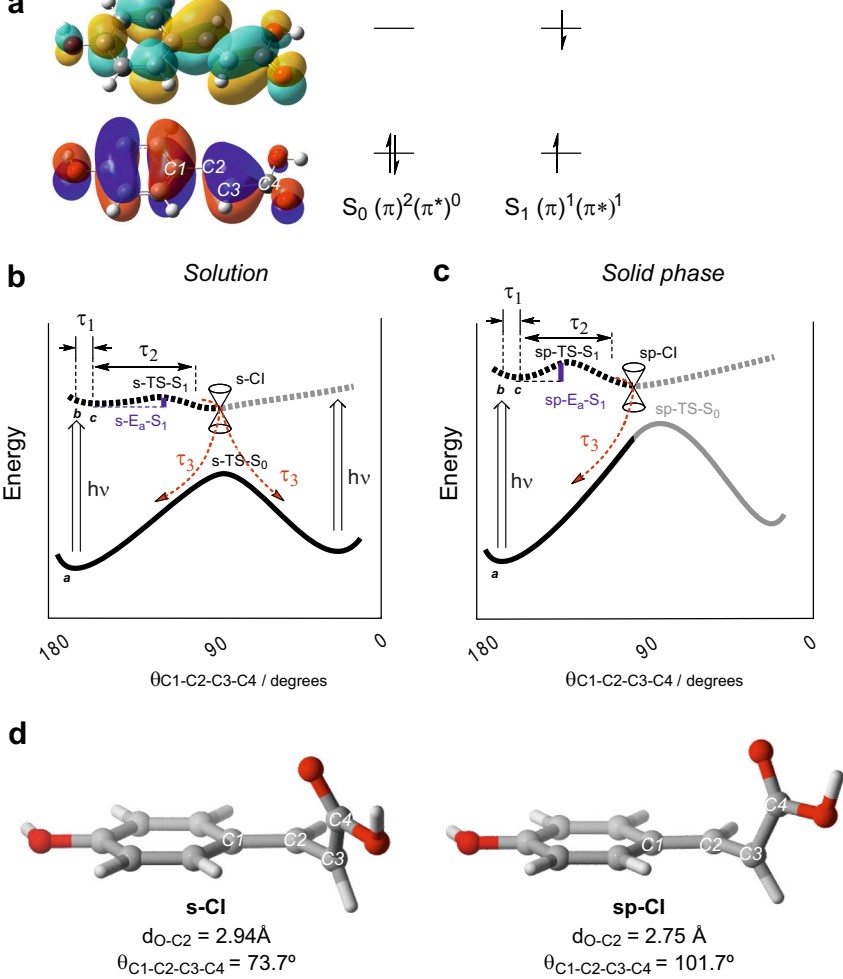

**Fig. 4 Energy profiles for the *trans-cis* isomerization in ground and excited electronic states. a** Frontier orbitals of *p*-coumaric acid and schematic representation of $S_0$ and $S_1$ electronic states computed at wb97XD/6-31 + G(d,p) level. **b** Solvated (s) model isomerization reaction profile computed at wb97XD (PCM, methanol)/6-31 + G(d,p) level. **c** Solid phase (sp) model isomerization reaction profile computed at wb97XD/6-31 + G(d,p) level. **d** Main geometrical features of conical intersections (CI) computed at CASSCF(12,11)/6-31 + G(d) (sp-CI) and CASSCF(12,11)/PCM,methanol/6-31 + G(d) (s-CI) levels.

back to the initial ground state, both isomers, *trans* and *cis*, could theoretically be formed. This is in accordance with the mixture of *trans* and *cis* isomers detected in the [1]H-NMR of *p*-coumaric acid solution after UV irradiation. Nevertheless, a different behaviour was obtained in a solid phase scenario (sp). In this case, an early CI $S_0$-$S_1$ (sp-CI) was obtained. This meant that the CI was reached before the rotation of the carboxyl group was completed ($\theta_{C1-C2-C3-C4}$ = 101.7°, Fig. 4d right)[47]. This implies that, once the molecule reaches the sp-CI-$S_0$, an additional amount of energy would be required to reach sp-TS-$S_0$ that is, the isomerization would be energetically unfavourable and, thus, the molecule returns to the initial *trans*-isomer geometry, again in agreement with the [1]H-NMR results after UV irradiation of powder *p*-coumaric acid, where the *cis*-isomer was not detected.

## Discussion

Phenolic acids present in the cuticle play a critical role in plant UV photoprotection, especially within the UV-C and UV-B range. Figure 5 depicts a schematic representation of the different UV protective mechanisms associated with the cuticle. From the results obtained with cuticles from leaves and fruits of different species, it can be inferred that the cuticle, as a continuous extracellular layer, provides an effective and spatially uniform UV-B protection to all plant tissues, including epidermal cells. This protection varies from almost complete to 80-90% for most of the studied cuticles. However, in species with a markedly low amount of cuticle, and consequently of cuticle phenolics, this UV-B attenuation was lower but still reached at least 50%. In these instances, intracellular accumulation of phenolic compounds, such as sinapates in *B. oleracea*[48], would notably contribute to UV-B screening. Reflectance played a minor yet variable contribution, 3–10% depending on the species, to this reduction of UV-B transmittance, whereas absorbance within the UV range was the main mechanism of UV-B light avoidance. It is important to mention that phenolic acids are also present in cell walls, especially in some plant families[49]. Hence, the non-cutinised section of the outer epidermal cell wall could act as a second and inner protective barrier against UV in those species with a particularly thin cuticle layer. In this regard, *B. vulgaris* cell walls contain ferulic acid esterified to pectic side chains[49]. Moreover, cell wall-bound phenolics, a combination of cuticle and cell walls, have been suggested to be responsible for UV tolerance differences in Antarctic mosses[50].

*Trans-cis* isomerization has been described as the mechanism for efficient energy dissipation in phenolic acids after UV absorption[22]. However, our results indicate that this isomerization

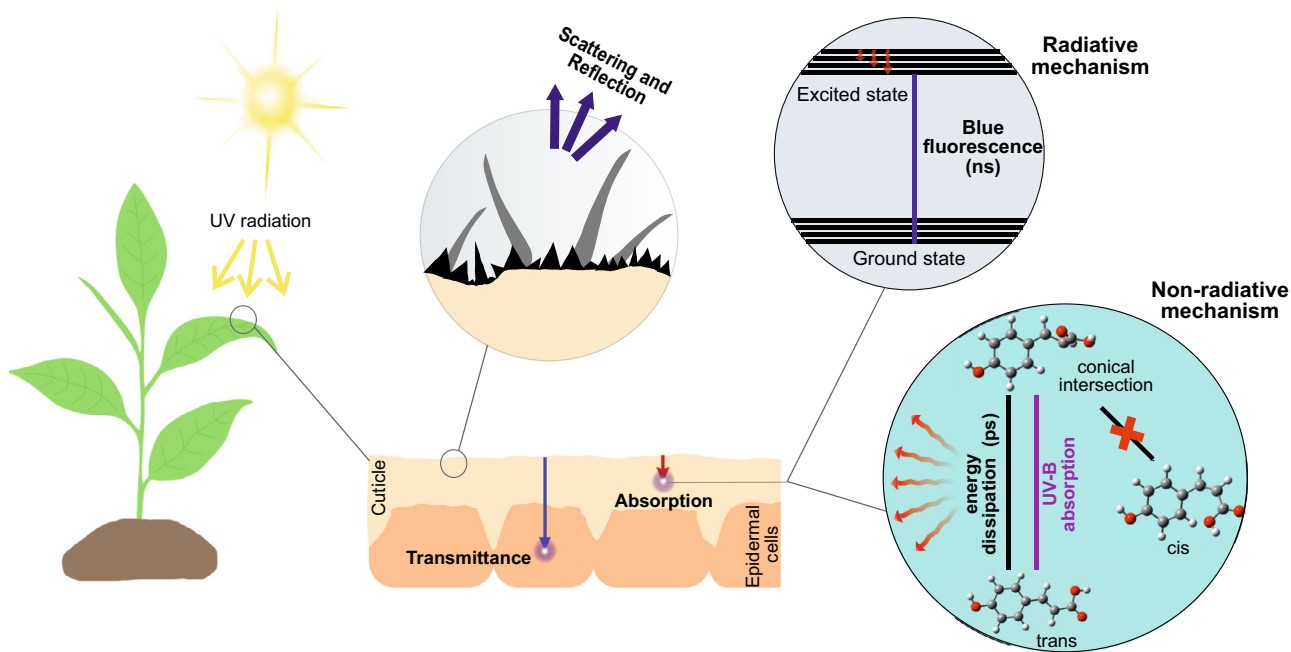

**Fig. 5 Schematics of the photoprotection processes observed in plant cuticles.** Of the UV light reaching the surface of the cuticle, a small fraction can be reflected or scattered due to the presence of trichomes, crystallised wax structures, etc. A variable fraction of UV light is transmitted to the epidermal cell. Within the UV-B range, transmittance varies among species from a negligible 0.3% to almost 50% in species with a very little amount of cuticle phenolics, such as *Brassica oleracea* and *Beta vulgaris*. Most of the UV light reaching the surface of the plant is thus absorbed by the phenolic compounds present in the cuticle. Part of this absorbed energy can be released with the emission of blueish fluorescence. However, most of the absorbed UV light seems to be dissipated via non-radiative mechanisms, being an ultrafast change in molecular geometry within the *trans* conformation of the phenolic acid and return to the ground state considered in the present work.

only occurs in a solvated environment, a likely approximation of the cellular cytoplasm or the vacuole. Thus, it would be expected that intracellular phenolic acids, that reportedly accumulate after UV exposure[6,7], could dissipate the absorbed energy via isomerization in those species where the cuticle considerably transmits UV light. However, in a solid matrix, either aggregated or chemically bonded, the combination of a constrained molecular scenario with the existence of an early $S_0$–$S_1$ conical intersection provides a non-radiative excited-state deactivation that does not imply any isomerization process. This unexpected absence of photoisomerization in a solid environment points towards a photodynamic behaviour based on a photostable non-radiative excited-state deactivation of phenolic acids. This ultrafast dissipative mechanism of UV energy, in combination with the much slower fluorescence emission and other non-radiative pathways such as vibrational relaxation, seems to act in combination to protect plant tissues from harmful UV radiation. As it was mentioned earlier, cinnamic acids exhibit important chemical features like photoprotection, photostability and antioxidant properties. Thus, their occurrence within the cuticle protects not only epidermal cells from damage to macromolecules, including DNA and proteins but also the cutin matrix itself from a potential lipid peroxidation due to their ROS scavenging activity. Additionally, it is worth mentioning that the conformational changes underwent by phenolic acids during the proposed mechanism of energy deactivation could imply a slight and fast rearrangement of the cutin matrix, which may be translated into dynamic fluctuations under natural conditions of some biophysical properties such as water uptake and rheology of the cuticle.

Differences in lifetimes and band shifts observed among the studied species could be attributed to the specific phenolic composition and the molecular and structural environment of their location within the cuticle. As it has been mentioned, phenolic acids have been reported in the epicuticular and intracuticular wax fraction, depending on the species, and esterified to the cutin matrix. In this sense, changes in matrix structure as the potential result of variation in cutin monomer composition and cross-linking could produce a more or less rigid matrix or with environments of different polarity that have been reported to modify and cause spectral modifications[22,23]. However, despite the differences, a robust and common mechanism of UV photoprotection can be observed in the cuticle of all the species studied which points to the idea of a conserved ancestral UV-screening mechanism. In the last years, an increasing number of cuticles from bryophytes and lycophytes are being studied and interestingly a notably high fraction of phenolics have been reported as an integral part of the cuticle in these species[18,19,51]. Moreover, cinnamic acids, specifically *p*-coumaric acid, have been identified in fossil plant cuticles[52]. Indeed, analysis of various bryophyte species showed that phenolics, mainly *p*-coumaric acid and to a lesser extent ferulic acid, constitute more than 50% of cutin monomers[18,19], whereas in seed plants seem to be minor constituents[19]. The presence of a cuticle highly enriched in cinnamic acids in bryophytes has been considered to play a role in cuticle impermeability[51] or defence against pathogens[52]. Here we posit that protection from UV damage whilst allowing visible radiation to reach the photosynthetic machinery, one of the main challenges for land inhabitation, could also be behind this accumulation. The presence of phenolic acids in the cuticle of species belonging to all the major plant groups points towards an ancestral mechanism of efficient energy dissipation. Furthermore, phenolic compounds have been identified in the macromolecular structure of other plant biopolymers, including lignin, suberin, sporopollenin as well as the cell wall[53,54]. Most of these biopolymers are located at or near the plant surface of aerial organs, and exposed to UV radiation; therefore, their potential photoprotective role, which could have been an adaptive strategy to

short wavelength radiation exposure, is a subject that needs further investigation.

## Methods

**Cuticle extraction and fresh tissue collection**. Leaf or fruit cuticles were enzymatically isolated from different plant species following the protocol established[55]. Briefly, tissues were incubated for at least 2 weeks in an aqueous solution of sodium citrate (50 mM, pH 3.7) with a mixture of fungal cellulase (0.2% w/v Sigma, USA) and pectinase (2.0% w/v Sigma, USA) together with 1 mM NaN$_3$ to prevent microbial growth. Once isolated from the epidermis, cuticles were incubated for another week in fresh enzymatic solution, then rinsed in distilled water and stored under dry conditions. Mature fruits were employed to isolate cuticles from *Capsicum annuum* L. and *Vitis vinifera* L., and fully expanded leaves to isolate adaxial cuticles of *Brassica oleracea* L., *Beta vulgaris* L., *Hedera helix* L., *Iris germanica* L., *Agave Americana* L. and *Clivia miniata* (Lindl.) Regel. To determine the micrograms of cuticle phenolics per surface area, cutin depolymerisation in methanol with 1% KOH was carried out for 24 h at 55 °C with continuous agitation and the absorbance of the solution measured in a UV-VIS spectrophotometer (Pharmacia Biotech, Piscataway, NJ, USA) at 324 nm. A calibration curve using a solution of naringenin dissolved in methanol with 1% KOH, following the protocol already established[56]. Three biological replicates from different plants were analysed.

Leaf samples were collected from *Adiantum raddianum* C.Presl, *Cycas revoluta* Thunb. and *Araucaria bidwillii* Hook. plants and kept in a moisturised environment until the spectroscopic analyses were performed. Cuticle and leaf samples were inspected under a stereomicroscope (100×, Leica, Heidelberg, Germany) to avoid regions with cracks or surface defects.

**Transmittance, absorbance and reflectance measurements**. Reflectance (%R) and transmittance (%T) spectra of plant cuticles were registered using a Cary 7000 UV-Vis-NIR spectrophotometer (Agilent Technologies, California) equipped with an external integrating sphere. Spectralon® masks of ~3 mm diameter were used to hold the cuticles during measurements. Spectra were recorded within the 200–800 nm range with a 1 nm resolution. Absorbance was calculated following the equation:

$$Abs = \log\frac{100 - \%R}{\%T}$$

where Abs is absorbance and %R and %T are the registered data of reflectance and transmittance, respectively. Four to six cuticle samples from different plants were analysed for each species. UV-Vis absorption spectrum of a $10^{-4}$ M methanolic solution of *p*-coumaric acid (≥98%, Sigma Aldrich, St. Louis, USA) was recorded at room temperature in a UV-1280 Multipurpose UV-Visible Spectrophotometer (Shimadzu, Kioto, Japan) within the 200 to 800 nm range in a quartz cuvette and with ~1 nm of spectral resolution.

Cuticle fluorescence emission spectra were recorded with an FLS920 spectrofluorometer (Edinburgh Instruments, UK) equipped with a 400 W Xe lamp for continuous measurements and with two photomultiplier detectors R2658P (200–1100 nm) and R928P (200–800 nm). Additionally, fluorescence lifetime measurements were carried out in the TCSPC fluorometer using an EPL-375 picosecond pulsed diode laser as an excitation source. Data were processed with the Edinburgh Instruments FAST software.

**Tissue sectioning and fluorescence microscopy**. Small pieces of fresh fruit or leaf samples were fixed in 4% paraformaldehyde, dehydrated in a series of ethanol and later on embedded in a commercial resin (Leica Historesin Embedding Kit, Heidelberg, Germany). Samples were cross-sectioned into slices 4 μm thick using a Leica microtome (RM2125; Heidelberg, Germany). Three biological samples were analysed per species. Samples were inspected under a fluorescence microscope (Nikon, Eclipse E800, Tokyo, Japan) using a UV filter (excitation filter: 330–380 nm; dichroic mirror:400 nm; barrier filter: 420 nm) and microphotographs taken with a Nikon camera (DXM1200) coupled to the microscope.

**Ultrafast transient absorption spectroscopy**. Time-resolved absorption spectra were collected using a broadband pump-probe transient absorption spectrometer (HELIOS, Ultrafast Systems, USA). Laser pulses (800 nm, 100 fs, 5 W) were produced by a regenerative Ti:Sapphire amplifier (Spitfire Ace, Spectra-Physics, USA) seeded with a Ti:Sapphire femtosecond laser (MaiTai SP, 80 mHz, Spectra-Physics, USA) and pumped by a Q-switched diode-pumped Nd:YLF laser (Empower 45, 1 kHz, Spectra-Physics, USA). This amplified laser pulse is split into two beams (Supplementary Fig. 5): one that passes through a tuneable optical parametric amplifier (TOPAS prime, Spectra-Physics, USA) that allows to select the excitation wavelength within the 290–1600 nm range (pump), and a second beam that goes to a variable mechanical delay line and generates a white light continuum by focusing into a CaF$_2$ crystal (probe). Both pulses (pump and probe) are aligned to overlap and reach the same sample spot. The pump pulse excites the sample and allows molecules to go from the ground to the excited state. Then, after the established delay time, the probe pulse reaches the sample and spectra of the excited molecule species are monitored before they return to the ground state, registering the excited-state processes and how they decay with time. Probe light transmitted

through the liquid sample (blue line in Supplementary Fig. 5) or reflected from the solid sample (yellow line in Supplementary Fig. 5) is collected by one mirror and, after passing several filters and additional mirrors, is detected by a CMOS (complementary metal-oxide-semiconductor) sensor. In the reflectance mode, there is an additional mirror and lens. This technique allows to register differences in absorption spectra with and without sample excitation.

Samples were photoexcited with 300 nm pump pulses, corresponding with the maximum UV absorption of *p*-coumaric acid and cuticles (Supplementary Fig. 1 and Fig. 1). Spectra were collected within a time range until 4 ns. Liquid samples of *p*-coumaric acid ($10^{-3}$ M in methanol) were placed in a quartz cuvette with continuous stirring and TAS spectra were registered with the transmittance mode. TAS spectra of isolated cuticles, leaf tissues, *p*-coumaric powder and a polyalkylcoumarate oligomer were registered using the reflection mode. Several measurements were taken on randomly chosen regions of each sample. A translating sample holder was employed to pseudo-randomly move the solid samples over the selected region and prevent photodegradation. Nevertheless, absorption spectra before and after TAS measurements were recorded in isolated cuticles of three different species displaying the variable amount of cuticle and phenolics to confirm this (Supplementary Fig. 9). Transient absorption spectroscopy (TAS) measurements showed no significant differences among samples and hence data were presented as an average of the three scans. The instrument response function (IRF) for the methanol solution was 194–233 fs. Pump fluence ranged between 4.4 and 11.3 mJ/cm$^2$, depending on the power employed. Power dependence studies were conducted for *p*-coumaric acid in methanol (Supplementary Fig. 10), powder *p*-coumaric and isolated cuticles of three different species (Supplementary Fig. 11) showing a one-photon dynamics for all the samples.

Prior to the sequential fitting analysis, spectra were cropped to 700 nm and scattered light and background subtracted using Surface Xplorer software (Helios, Ultrafast systems, USA). A sequential fitting procedure[57] was employed using Glotaran software[58] to quantitatively determine the molecular dynamic processes and lifetimes observed in TAS. Data were sequentially fitted to obtain the fitted lifetimes (τ) and their associated errors as well as the evolution-associated decay spectra (EADS). The goodness of fit can be estimated from the residuals (Supplementary Figs. 12, 13, 14) and the comparison between experimental and fitted time traces (Supplementary Figs.15, 16, 17). Prior to the fitting, the dispersion was corrected with Glotaran software.

**$^1$H-Nuclear magnetic resonance (NMR) measurements**. $^1$H-NMR (400 MHz, DMSO-D6) spectra of *p*-coumaric acid in solution and powder were recorded using NMR Spectrometer Bruker Avance III 400 after irradiation at 300 nm for 5 h with a TOPAS amplifier (Spectra-Physics, USA) using a pump beam fluence of 0.1 mJ cm$^2$. Additionally, $^1$H-$^1$H correlation spectroscopy (COSY) was carried out to confirm the presence of the *cis*-isomer in the *p*-coumaric solution (Supplementary Fig. 19).

**Computational methodology**. Computational studies were carried out with Gaussian 16[59] (optimisations and conical intersections) or OpenMolcas[60] (CASPT2 single point energy calculations) software. Optimisation of stationary points in S$_0$, S$_1$ and T$_1$ electronic states were performed within the time-dependent–density functional theory (TD–DFT) framework[41] using the wb97XD[61] functional in combination with 6-31 + G(d,p) basis set. This range–separated hybrid functional includes Grimme's D2 dispersion correction[62] and has been reported to be reliable for an accurate description of excited states in conjugated organic compounds[63]. Solvent effects were estimated using the polarisable continuum model[64] (PCM) within the self-consistent reaction field (SCRF) approach[65] using methanol (ε = 32.613) as model solvent. Similarly, gas-phase calculations were used as a minimalistic model for solid-state analysis[66]. This approach has been reported for the analysis of the intrinsic electronic properties of *p*-coumaric acid (i.e. unaffected by external factors such as solvation)[47]. Dynamic electron correlation has been considered by means of CASPT2(PCM)/cc-pVTZ or CASPT2/cc-pVTZ single point energy calculations.

All the stationary points were characterised by harmonic vibrational analysis. Reactants, intermediates and products showed positive definite Hessians. Transition structures (TSs) showed only one imaginary frequency associated with nuclear motion along the chemical transformation under study. Activation and reaction (Gibbs) energies were calculated at 298.15 K considering species directly connected to the computed transition structures, both in the gas phase and in solvated conditions.

Conical intersections were carried out by means of Complete Active Space–Self-Consistent Field (CASSCF)[42] calculations using 6-31 + G(d, p) basis set. We selected the five occupied π–symmetry orbitals, the occupied non–bonding orbital of oxygen atom in the carbonyl group, and the five unoccupied π–symmetry orbitals as active space for a proper description of the system (namely CASSCF (12, 11) method)[45] starting from Hartree–Fock initial Hessian (see Supplementary Fig. 18 for more details on the chosen active space). CASPT2 energy calculations were performed by using the same CASSCF (12, 11) active space as a reference state. Cartesian coordinates of all stationary points and conical intersections are presented in Supplementary Fig. 20.

**Statistics**. Curvilinear regression was employed to study the relationship between cuticle absorbance and the amount of phenolics. Linear regression was also employed for power dependence studies. All regressions were statistically significant with $p < 0.05$. SPSS software[67] and Origin(Pro) (OriginLab Corporation, MA, USA) were used for the analyses.

**Reporting summary**. Further information on research design is available in the Nature Research Reporting Summary linked to this article.

## Data availability

Source data are provided with this paper.

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

## Acknowledgements

This work has been supported by grants RTI2018-094277-B/AEI/10.13039/501100011033, PID2020-119636GB-I00 and RED2018-102331-T from Agencia Estatal de Investigación, Ministerio de Ciencia e Innovación, Spain co-financed by the European Regional Development Fund (ERDF) and SBPLY/17/180501/000189 from UCLM Junta de Comunidades de Castilla-La Mancha (JCCM-FEDER). Computational resources, technical and human support provided by IZO-SGI, SGIker (UPV/EHU, MICINN, GV/EJ, ERDF and ESF) and DIPC (Donostia International Physics Centre) are gratefully acknowledged. Ana González Moreno is the recipient of a Formación Personal Universitario fellowship FPU17/01771. The authors thank Dr. Jessica Román, Dr. José Luis Zafra and Dr. Cristina Capel for their technical work and Dr. Victoria Gómez, Universidad de Castilla-La Mancha, for the NMR assignments.

## Author contributions

A.G.M., A.d.C., and P.P. performed TD–DFT calculations. A.G.M. carried out TAS data analysis, sequential fitting calculations, analyses of the optical properties and contributed to data interpretation. A.d.C. performed CASSCF and CASPT2 calculations and wrote the theoretical calculations section. E.D. performed sample selection and collection, cuticle isolation and microscopy studies. A.H. and E.D. conceptualised the work. A.H. compiled all the results and wrote the first draft of the manuscript. All authors contributed to writing the manuscript and agreed on the final version.

## Competing interests

The authors declare no competing interests.
