## [Peer Review File · Nature Communications]

Radiationless mechanism of UV deactivation by cuticle phenolics in plantsREVIEWER COMMENTS

Reviewer #1 (Remarks to the Author):

The Manuscript by Ana González Moreno, et al., focuses on the photochemistry of hydroxycinnamic acids in a range of different plant cuticles. The photodynamics and their associated timescales are acquired using both reflective and transmissive femtosecond transient spectroscopy. These measurements are supported by additional techniques such as NMR and UV/vis spectra, and complimentary ab initio calculations to aid analysis. Studying the photodynamics of the hydroxycinnamic acids in the plant cuticle environment is novel allowing for modelling them in their real environment. This highlighted that the photodynamics had a species dependence. The data present is of good quality, however, the analysis used to extract the dynamics and time-constants is done via a global fit using concurrent exponential fits, rather than a sequential fit. I suggest that the authors reanalyse their TAS data using a sequential model. Due to this, whilst the paper is of interest to a range of audiences and contains a novel approach to studying the photodynamics of hydroxycinnamic acids in a realistic environment I cannot recommend this paper for publication without addressing several issues; such as the reanalyse of the TAS data via a sequential fit.

Comments:

The authors attribute τ_3 to the vibrational relaxation to the solvent of the vibrationally hot ground state for coumaric acid, however this time-constant has previously been shown to be isomerization on the excited state, especially since stimulated emission is usually present in a polar environment, such as methanol. I believe that this is not observed in the TAS of the authors due to the appearance of the phenolic radical of coumaric acid. The presence of the phenolic radical could be confirmed by power dependency studies on the features in the TAS. I recommend that the authors perform power dependency studies on all their measurements in order to confirm they are seeing dynamics due to one-photon absorption rather than multi-photon absorption.

Furthermore, regarding power dependency studies, the authors have not included their pump fluence they used for the TAS measurements. Can they also provide a value for their instrument response function at sample?

During the measurements of the cuticles, was any translation performed to avoid any photodegradation of the sample? Or was randomisation the of the time-delay ordering during each average used, to reduce the impact of any sample degradation? The authors could confirm how much degradation (if any) the sample underwent by doing a prolonged UV irradiation study on the cuticles, by taking a UV/vis spectrum before and after irradiation.

Did the authors carry out the TAS measurements with the WLC and pump polarization at magic angle?
As no wave-plate is shown on their experimental diagram.

I assume the authors meant Lens instead of Lent in the diagram caption.

Reviewer #2 (Remarks to the Author):

The manuscript reports on studies of the function of hydroxycinnamic acids present in plant cuticles. To this purpose femtosecond transient UV-Vis absorption spectroscopy has been performed on solid biological samples, as well as on the solvated chromophore. Combined with quantum chemical calculations these studies have led the authors to conclude that cuticle phenolics serve as a photoprotection barrier from harmful UV radiation.

I am not a biologist and can therefore not judge what the impact will be in the biological/plant community of the conclusion regarding the photoprotective function of cuticles. From a spectroscopic point of view, however, the observations and conclusions drawn by the authors do not come as a big surprise. In recent years there has been an steeply increasing interest in artificial and biological sunscreen filters, with phenolic compounds based on cinnamic acids being one of the classes of chromophores that are receiving particular attention. The present studies follow to a large extent what is known from these studies, albeit that studies on solid samples are rare and from that respect provide indeed novel information. Spectroscopically the work thus does not meet the stringent conditions on novelty and impact as imposed by Nature Communications, and from that point of view I can therefore not recommend publication. Apart from these general considerations, there are also a number of more detailed issues that need further attention (not necessarily in order of importance).

1. Supplementary Fig. 2 reports autofluorescence of plant cuticles. Apart from the fact that it is very hard to distinguish the emission from the background (it should be possible to present the Figure in another way), it would be useful to characterize this emission in more detail by recording dispersed emission and fluorescence excitation spectra, as well as determining fluorescence lifetimes. The former should have the signatures of the phenolics spectra, while the latter -if the assignment of the authors of $\tau(4)$ to emissive decay is correct (which I doubt) should reproduce these $\tau(4)$ lifetimes.

2. Figure 2a reports transient absorption spectra of p-coumaric acid (pCA) in methanol. These spectra differ considerably from spectra that in the past have been measured in buffered solutions (J. Phys. Chem. B 109, 4197-4208 (2005)). In particular, previously a clear stimulated emission (SE) band was observed that is absent in the present spectra. It is not clear why there would be such large differences. Moreover, the present spectra show an induced absorption (IA) band around 500-700 nm that is attributed to "charge transfer reactions within the molecule or with the solvent". It is not clear to me

what reactions the authors refer to, also because no reference to previous literature is given. If they intend bond breaking of the OH, which would lead to a phenolic radical and a solvated electron, one would expect to see the characteristic features of these two species but these are absent. Another problem is that if other species would be created one would expect to see distinct decay (or recovery) dynamics of these species.

3. Global fitting has been used to analyse the TA spectra and is reported in Supplementary Figures 4, 5 and 7. The upper panels in these Figures display differences between measured and fitted data, but are completely illegible, showing merely dark blue panels.

4. Table 4 reports the fitted lifetimes. While I can agree with the interpretation of $\tau(2)$ and $\tau(3)$ for pCA in solution, which follows previous conclusions, I have more problems with their interpretation in pCA powder and oligomer. Studies reported in PCCP 19, 21127-21131 (2017) in which solvent viscosity effects on photoisomerization of sinapoyl malate are studied, and in J. Lumin. 206, 469-473 (2019) where excited-state dynamics of sinapate esters in solution and polymer films are compared clearly indicate that in pCA powder and oligomer $\tau(2)$ is associated with vibrational cooling in the electronically excited state and $\tau(3)$ with internal conversion to the ground state along a trans-cis photoisomerization coordinate. Such an assignment would in fact nicely follow the discussion of the authors on their computational results.

5. Invoking a 'lifetime' $\tau(4)$ is too suggestive. As far as I can see from Supplementary Fig. 3 and Figs. 2 and 3 the maximum delay of the TAS setup is 2 ns. The presence of a third component in the global analysis thus merely indicates that there is a long-lived species with a lifetime beyond a few nanoseconds or so, but this species would not necessarily need to be an electronically-excited state species but could as well be a ground-state species such as the cis-isomer.

6. I similarly have problems with the interpretation of $\tau(4)$ as being associated with an additional relaxation mechanism based on excited-state emission. This would imply that population would get trapped in a long-lived emissive state, but the calculations reported here nor computational studies of similar compounds do not give an indication for the presence of such a state. Moreover, also the measured TAS do not give indications for such emissions. There is, on the other hand, extensive literature on gas-phase studies of cinnamates, coumarates, ferrulates, and sinapates by Buma et al. and Ebata et al. that demonstrate the important role of ISC. This pertains in particular to coumarates for which a low-lying $n\pi^*$ state allows for an efficient ISC to the triplet manifold and decay of the thus populated lowest $T_1(\pi\pi^*)$ triplet state that decays on a tens of ns time scale (see ref. 29 cited by the authors for the pertaining discussion on how this is manifested there in the solution excited-state dynamics). Such pathway definitely also need to be considered in the present studies.

7. Supplementary Figures 4, 5, and 7 report decay associated spectra (DAS), but only for $\tau(2)$ and $\tau(3)$. The authors claim $\tau(1)$ cannot accurately be resolved, but in view of the characteristics of their laser system (100 fs pulses) and previously reported decay times (of the order of 100 fs) it should be possible to say something -albeit not so accurately- about this decay time. Reporting of the DAS associated with $\tau(1)$ and $\tau(4)$ would be useful, but even more useful would be reporting of evolution associated difference spectra (EADS). The authors focus their discussion primarily on the decay times, but hardly consider the spectral evolution of their TAS, while this spectral evolution definitely contains a lot of information on the different steps of the excited-state decay process.

8. In order to elucidate the observed excited-state dynamics and the differences between a solution and solid state environment the authors report quantum chemical calculations in a solvated environment and in a minimalistic model of the solid environment. Although for the latter is referred to the Methods section, this section does not contain any description of this minimalistic model and can therefore not be assessed on its validity. CASSCF calculations have been performed with an active space of four electrons and six orbitals. In view of the experimental observations that the $n\pi^*$ state plays an important role in the $\pi\pi^*$ excited state dynamics as well as the results of previous theoretical work (Chem. Phys. 313, 71-75 (2005); Spectrochim. Acta Part A 211, 203-211 (2019)) that use spaces of (12,11) and (10,8), respectively it is doubtful whether the (4,6) active space is large enough. Also, dynamic electron correlation has for this chromophore been shown to be important so CASPT2 single point calculations would be quite useful. The authors remark that in order to avoid inconsistencies S_0 and S_1 energies at conical intersection geometries were recalculated by means of TD-DFT calculations, but the usefulness of these energies is very doubtful in view of the inadequacy of DFT calculations to describe CIs.

Reviewer #3 (Remarks to the Author):

The article entitled "Radiationless mechanism of UV deactivation by cuticle phenolics in plants" by González Moreno et al. is a study of the interactions between light and phenolics (specifically cinnamic acid-type molecules) in a plant cuticle or cuticle-like environment. In particular, the authors investigate the potential of phenolics to act as barriers to UV light. For this, they examine the transmittance, absorbance, and reflectance properties of plant cuticles with varying phenolic content (Fig. 1), the absorption of the phenolic compound p-coumaric acid in a time-resolved manner in both solution and solid-state configurations (picosecond scale, Fig. 2) and the time-resolved absorption characteristics of isolated plant cuticles (Fig. 3). Finally, the authors use quantum chemical calculations to determine energy profiles for p-coumaric acid during cis-trans isomerization (Fig. 4). Considering the data together, the authors propose (i) that plant cuticles can absorb UV light, (ii) that cuticle UV absorption is associated with the abundance of phenolic compounds, and (iii) that the dissipation of absorbed energy does not occur via cis-trans isomerization but instead by a combination of other mechanisms, both radiative (fluorescence) and non-radiative (rapid shifts in molecular geometry). The data appear to be sound. I believe the topic is interesting and appropriate in scope for Nature Communications. I have several recommendations, both major and minor for how the article could be improved, and list these below.

Major comments

In the first section of the results, the authors seem to be trying to correlate the abundance of phenolics in the cuticle with absorbance in the UV-B range. If this is what the authors are wishing to do, then they should create a scatter plot and regress the abundance of cuticle phenolics against absorbance. With the data underlying that plot they can then also apply regression statistics to determine if the relationship they propose is statistically significant. Including actual regression statistics will enhance the quality of the analyses, and, if the correlation is significant, strengthen the authors arguments.

The authors nicely show that some of the light absorbed by cuticle phenolics is dissipated by fluorescence. They also provide good evidence that cis-trans isomerization is not occurring in the solid state (i.e. in the cuticle) but that changes in molecular geometry are still occurring that could enable energy dissipation. However, I see rather little evidence for the relative contributions of these two mechanisms to the overall dissipation of the absorbed UV energy – just the statement about quantum yield (lines 158-159). The authors state in several places that the non-radiative mechanism predominates. What is the rationale for those statements? The manuscript should be modified to emphasize and make that rationale very clear.

Minor comments

Lines 140: “Since none of the aliphatic and terpenoid compounds of waxes [...] have conjugated double bonds”. This is true for some species, but not all. Alkyl resorcinols, found in a variety of plant species wax mixtures, contain conjugated double bonds. Please modify the sentence to make it more accurate.

The article needs to be thoroughly proofread. Many errors, including some that should have been caught by an automated spell-checking tool, can be found throughout the manuscript. I list a few here, but the list is non-exhaustive:

Line 85: “present in the cuticle have already shown to play...” -> “present in the cuticle have already been shown to play...”

Line 176: “p-coumaric was also analyzed” -> “p-coumaric acid was also analyzed”

Line 224: “corresponded to the cis-isomer However” -> “corresponded to the cis-isomer. However” (missing period)

Phenolic acids present in the cell walls, which the authors suggest may absorb UV light, are not present in Fig. 5. The authors should consider adding them to make the figure more comprehensive.

Line 70/71: “Waxes can have a dual location, either vanishing the surface (epicuticular...”, what does “vanishing” mean here? Do the authors mean that the waxes vanish to the surface? If so, I am not sure

vanishing is the right word, since waxes are particularly visible when on the surface. Perhaps change this word to help improve reader comprehension.

Line 51: “cope” -> “overcome”

Quite a few places are missing citations. Please search the manuscript for all such instances and correct this. Some starting points are: (i) lines 76-78, ending in “acting a thermal regulator”, (ii) lines 148-150, “Of the different phenolics present in plant cuticles, p-coumaric acid has been identified ubiquitously and in some instances as the only cinnamic acid derivative.”, and (iii) lines 126-128, “Epicuticular wax disposition and crystallization, as well as the presence and density of indument, are chiefly responsible for light reflection and could be behind the observed differences among species.”

Line 112: “reduced cuticles”. Do the authors mean reduced abundances of phenolics? “Reduced cuticles” could refer to wax, cutin, phenolics, or all the above. Please be more specific.

While reading the first section of the results section, the first thing I thought of was “what does the absorption spectrum of a p-coumaric acid standard look like?”. Would it make more sense to introduce the absorption characteristics of the p-coumaric acid standard at the beginning of the first results section, rather than at the end of that section? Just an idea.

Line 375-378 “This ultrafast dissipative mechanism of UV energy, in combination with the much slower fluorescence emission and other non-radiative pathways such as vibrational relaxation act in combination to protect plant tissues from harmful UV radiation.” This is a rather strong statement that could, in the future, be rendered incomplete or even inaccurate, for example, if we were to learn more about the absorption properties of phenolics inside the cuticle other than p-coumaric acid. Perhaps “seem to act in combination” would be a safer and more appropriate configuration for this sentence.

Line 401: “basal plants”. In this context, what does “basal” mean? Does it mean that the branches of the plant phylogeny leading to bryophytes are at the bottom of the tree? If this is what is meant, then that is inaccurate (see: McDaniel, S.F., 2021. Bryophytes are not early diverging land plants. *New Phytologist*, 230(4), pp.1300-1304.). The argument the authors appear to be making is that phenolics are found in diverse land plants. If this is the case, then I think substituting the word “basal” for the word “diverse” will both strengthen the authors argument and make the sentence more accurate.

REVIEWER 1

The data present is of good quality, however, the analysis used to extract the dynamics and time-constants is done via a global fit using concurrent exponential fits, rather than a sequential fit. I suggest that the authors reanalyse their TAS data using a sequential model.

We have carried out a sequential fit of all the TAS presented and the results are shown in the new Table 1 and the software employed is indicated in the materials and methods section (see lines 523-527). Although the lifetimes are different from those obtained with the global analysis, they remain in the same range and displaying the same behaviour.

The authors attribute tau3 to the vibrational relaxation to the solvent of the vibrationally hot ground state for coumaric acid, however this time-constant has previously been shown to be isomerization on the excited state, especially since stimulated emission is usually present in a polar environment, such as methanol. I believe that this is not observed in the TAS of the authors due to the appearance of the phenolic radical of coumaric acid. The presence of the phenolic radical could be confirmed by power dependency studies on the features in the TAS. I recommend that the authors perform power dependency studies on all their measurements in order to confirm they are seeing dynamics due to one-photon absorption rather than multi-photon absorption.

The referee is right, tau3 can be assigned to the isomerization process in the excited state. The ms has been modified accordingly (see lines 203-204 and Fig 4).

Regarding the stimulated emission, in the previous version of the ms the TAS of coumaric acid was carried out in basic methanol. We have carried out TAS measurement of coumaric acid in methanol and at a higher concentration and the stimulated emission is observed (see Fig 2a and lines 174-175).

Regarding the power dependence studies, they have been carried out for coumaric acid in methanol (see suppl Fig 13) and for coumaric powder and three cuticles (see Suppl Fig 14). Results indicate a one-photon absorption. The ms has been modified to include this information (see lines 516-519).

Furthermore, regarding power dependency studies, the authors have not included their pump fluence they used for the TAS measurements. Can they also provide a value for their instrument response function at sample?

The pump fluence used in the TAS measurements has been included (see lines 514-516).

The instrument response function (IRF) for methanol was carried out and the value included in the ms (see lines 513-514). However, it has not been done with cuticles since it would imply having cuticles without phenolics, and that is not possible. In order to remove the phenolic fraction of the cuticle, the cutin matrix (the main component of the cuticle, 60-80% of the cuticle depending on the species) needs to be depolymerized (see ref 56 in the ms), hence rendering impossible to obtain a cuticle without phenolics for each species. For similar reasons it could not be performed an IRF analysis for leaf epidermises and the other solids. The sequential analyses performed with Glotaran software give an approximate IRF for the solid samples that ranged between 400 and 700 fs. However, for the coumaric acid in methanol the Glotaran IRF was lower than the calculated IRF for methanol, thus we believe this is also true for the solid samples.

During the measurements of the cuticles, was any translation performed to avoid any photodegradation of the sample? Or was randomisation the of the time-delay ordering during each average used, to reduce the impact of any sample degradation? The authors could confirm how much degradation (if any) the sample underwent by doing a prolonged UV irradiation study on the cuticles, by taking a UV/vis spectrum before and after irradiation.

A translating sample holder was employed to randomly move the cuticle sample over the selected area. The methods section has been modified to specify this point (see lines 506-508). To confirm that no photodegradation of cuticles took place during TAS, an UV/vis absorption spectra was taken before and after a TAS experiment, as the referee suggested, for three different cuticles displaying different thicknesses and amount of phenolics (see Suppl Fig 12 and lines 508-511). The result showed that the spectra were the same, without any qualitative or quantitative differences, thus indicating that no photodegradation had taken place during TAS analyses.

Did the authors carry out the TAS measurements with the WLC and pump polarization at magic angle? As no wave-plate is shown on their experimental diagram.

TAS measurements were carried out with a WLC, as indicated in line 486. In the standard configuration of the Helios Ultrafast system a broadband depolarizer for the pump is used. In this way we don't have to worry about the magic angle. The achromatic depolarizer used is either DPU25 with -A coating from ThorLabs or DPP-25-A (specified wavelength range is 350-700 nm).

I assume the authors meant Lens instead of Lent in the diagram caption.

Yes, it has been modified (see Suppl Fig 5).

REVIEWER 2

The present studies follow to a large extent what is known from these studies, albeit that studies on solid samples are rare and from that respect provide indeed novel information. Spectroscopically the work thus does not meet the stringent conditions on novelty and impact as imposed by Nature Communications, and from that point of view I can therefore not recommend publication.

As the referee indicates there is a good number of studies on photoprotection of cinnamic acids derivatives. Most of them were made on these molecules in solution. They have been considered as a source of inspiration. In our ms the study has been focused on isolated cuticles (solid biological samples) containing one of these cinnamic acids (*p*-coumaric) at low amounts and with a potential critical role in plant photoprotection against UV radiation that has never been investigated in detail. Most of the research on this subject in the plant physiology and biophysics fields take into account the presence of phenolics within cells but never at the outermost cuticle level.

1. Supplementary Fig. 2 reports autofluorescence of plant cuticles. Apart from the fact that it is very hard to distinguish the emission from the background (it should be possible to present the Figure in another way), it would be useful to characterize this emission in more detail by recording dispersed emission and fluorescence excitation spectra, as well as determining fluorescence lifetimes. The former should have the signatures of the phenolics spectra, while the latter -if the assignment of the authors of tau(4) to emissive decay is correct (which I doubt) should reproduce these tau(4) lifetimes.

Autofluorescence of epidermal and epicarp sections were presented to show the low yet cuticle localized blue fluorescence. From a plant biology perspective, it is important to show that the autofluorescence is located to the cuticle region and that underneath tissues do not exhibit (or exhibit comparatively lower) fluorescence. For comparison purposes, all images were taken under the same conditions of exposure etc. To better show this autofluorescence we have recorded, as the referee suggested, the fluorescence emission spectra (see Suppl Fig 4 and lines 157-160). Fluorescence lifetimes for the cuticles ranged between 2 and 8 ns.

Regarding the assignation of tau4 to fluorescence we agree with the referee and this has been modified in the ms following some of the referee's comments in points 5 and 6 (see lines 203-205).

2. *Figure 2a reports transient absorption spectra of p-coumaric acid (pCA) in methanol. These spectra differ considerably from spectra that in the past have been measured in buffered solutions (J. Phys. Chem. B 109, 4197-4208 (2005)). In particular, previously a clear stimulated emission (SE) band was observed that is absent in the present spectra. It is not clear why there would be such large differences. Moreover, the present spectra show an induced absorption (IA) band around 500-700 nm that is attributed to charge transfer reactions within the molecule or with the solvent. It is not clear to me what reactions the authors refer to, also because no reference to previous literature is given. If they intend bond breaking of the OH, which would lead to a phenolic radical and a solvated electron, one would expect to see the characteristic features of these two species but these are absent. Another problem is that if other species would be created one would expect to see distinct decay (or recovery) dynamics of these species.*

We thank the referee for this criticism. We have carried out a TAS of coumaric acid in methanol at a higher concentration and the stimulated emission was observed (see Fig 2a and lines 174-175)., at the same time the 500-700 nm did not appear and it was removed from the ms.

3. *Global fitting has been used to analyse the TA spectra and is reported in Supplementary Figures 4, 5 and 7. The upper panels in these Figures display differences between measured and fitted data, but are completely illegible, showing merely dark blue panels.*

After reanalysis of the TAS using a sequential fit, as indicated by referee 1, the corresponding residual heat maps have been zoomed in to better display the differences between the observed data and model prediction. The upper limit of the time axes has been set to include tau3 (see Suppl Figs 6, 7 and 9).

4. *Table 4 reports the fitted lifetimes. While I can agree with the interpretation of tau(2) and tau(3) for pCA in solution, which follows previous conclusions, I have more problems with their interpretation in pCA powder and oligomer. Studies reported in PCCP 19, 21127-21131 (2017) in which solvent viscosity effects on photoisomerization of sinapoyl malate are studied, and in J. Lumin. 206, 469-473 (2019) where excited-state dynamics of sinapate esters in solution and polymer films are compared clearly indicate that in pCA powder and oligomer tau(2) is associated with vibrational cooling in the electronically excited state and tau(3) with internal conversion to the ground state along a trans-cis photoisomerization coordinate. Such an assignment would in fact nicely follow the discussion of the authors on their computational results.*

The assignments of tau2 and tau3 have been modified according to the referee's indication and also following referee's 1 comments (see lines 200-203). Also, we thank the referee for the references indicated that support the interpretation of solid samples. They have been included in the ms (see refs 36 and 37 and lines 214-215).

5. Invoking a lifetime tau(4) is too suggestive. As far as I can see from Supplementary Fig. 3 and Figs. 2 and 3 the maximum delay of the TAS setup is 2 ns. The presence of a third component in the global analysis thus merely indicates that there is a long-lived species with a lifetime beyond a few nanoseconds or so, but this species would not necessarily need to be an electronically-excited state species but could as well be a ground-state species such as the cis-isomer.

As we have previously indicated in point 1, the assignment of tau4 to fluorescence has been removed from the ms and modified following the referee's suggestion (see lines 203-205).

6. I similarly have problems with the interpretation of tau(4) as being associated with an additional relaxation mechanism based on excited-state emission. This would imply that population would get trapped in a long-lived emissive state, but the calculations reported here nor computational studies of similar compounds do not give an indication for the presence of such a state. Moreover, also the measured TAS do not give indications for such emissions. There is, on the other hand, extensive literature on gas-phase studies of cinnamates, coumarates, ferrulates, and sinnapates by Buma et al. and Ebata et al. that demonstrate the important role of ISC. This pertains in particular to coumarates for which a low-lying $n\pi^$ state allows for an efficient ISC to the triplet manifold and decay of the thus populated lowest $T_1(\pi\pi^*)$ triplet state that decays on a tens of ns time scale (see ref. 29 cited by the authors for the pertaining discussion on how this is manifested there in the solution excited-state dynamics). Such pathway definitely also need to be considered in the present studies.*

Concerning the exact assignment of tau4 and following the comments of the referee in the point 1 we have indicated the this time could be assigned to a long lived state molecular state.

7. Supplementary Figures 4, 5, and 7 report decay associated spectra (DAS), but only for tau(2) and tau(3). The authors claim tau(1) cannot accurately be resolved, but in view of the characteristics of their laser system (100 fs pulses)

and previously reported decay times (of the order of 100 fs) it should be possible to say something -albeit not so accurately- about this decay time. Reporting of the DAS associated with $\tau(1)$ and $\tau(4)$ would be useful, but even more useful would be reporting of evolution associated difference spectra (EADS). The authors focus their discussion primarily on the decay times, but hardly consider the spectral evolution of their TAS, while this spectral evolution definitely contains a lot of information on the different steps of the excited-state decay process.

EADS spectra have been incorporated to the new Suppl Figs. 6, 7 and 9 and indicated in the ms following the referee's suggestion. IRF calculation of methanol, carried out following referee's 1 indication, showed a 194-233 fs value (lines 513-514), thus τ_1 must be less than 200 fs.

8. In order to elucidate the observed excited-state dynamics and the differences between a solution and solid state environment the authors report quantum chemical calculations in a solvated environment and in a minimalistic model of the solid environment. Although for the latter is referred to the Methods section, this section does not contain any description of this minimalistic model and can therefore not be assessed on its validity. CASSCF calculations have been performed with an active space of four electrons and six orbitals. In view of the experimental observations that the $n\pi^$ state plays an important role in the $\pi\pi^*$ excited state dynamics as well as the results of previous theoretical work (Chem. Phys. 313, 71-75 (2005); Spectrochim. Acta Part A 211, 203-211 (2019)) that use spaces of (12,11) and (10,8), respectively it is doubtful whether the (4,6) active space is large enough. Also, dynamic electron correlation has for this chromophore been shown to be important so CASPT2 single point calculations would be quite useful. The authors remark that in order to avoid inconsistencies S_0 and S_1 energies at conical intersection geometries were recalculated by means of TD-DFT calculations, but the usefulness of these energies is very doubtful in view of the inadequacy of DFT calculations to describe CIs.*

We thank the reviewer for these constructive comments.

We have included in the Methods section of the ms the description of the use of solid-state minimalistic model (lines 546–550) including a reference 65 in which are included the strong points and weakness of this approach. Moreover, an example in which the authors use gas phase calculations for the analysis *p*-coumaric acid (reference 47) is included.

We agree with the reviewer's comment about the limited reliability of the chosen active space in the initial version of the ms. To solve this issue, new CIs have been computed at CASSCF(12,11) level considering the π -symmetry set of

occupied and unoccupied orbitals and the n orbital of oxygen atom as indicated by the reviewer's comments and the literature. The ms has been adapted to include the new CASSCF(12,11) results in the Mechanism of phenolic UV deactivation section, Fig 4 has also been modified, and the new active space used for the calculations has been specified in the Methods section (lines 560–583) and in the Supplementary Figure 15.

Unfortunately, we do not currently have the required software to carry out calculations employing CASPT2 method for considering the dynamic electron correlation. However, we used an available alternative performing single point energy using Moller-Plesset perturbation theory MP2 method as implemented in Gaussian16 package to take this factor into account. The ms has been adapted to include the dynamic electron correlation in the analysis of the energetic profiles (lines 305–311). We found small deviations in the S_0 - S_1 energetic differences when using CASSCF(12,11) or MP2 energy values. Therefore, additional conical intersection geometries have been collected in Supplementary Table 3. Some comments have been included in the Supplementary information to clarify this point.

The MP2 energy values of all stationary points have been included in the Supplementary information (Supplementary tables 1-3), and Supplementary Figure 10 has been modified accordingly.

REVIEWER 3

Major comments

In the first section of the results, the authors seem to be trying to correlate the abundance of phenolics in the cuticle with absorbance in the UV-B range. If this is what the authors are wishing to do, then they should create a scatter plot and regress the abundance of cuticle phenolics against absorbance. With the data underlying that plot they can then also apply regression statistics to determine if the relationship they propose is statistically significant. Including actual regression statistics will enhance the quality of the analyses, and, if the correlation is significant, strengthen the authors arguments.

We thank the referee for this suggestion. The plot has been included together with the non-linear regression analysis in Supplementary Fig 2 and mentioned in the ms (see lines 149-152)

The authors nicely show that some of the light absorbed by cuticle phenolics is dissipated by fluorescence. They also provide good evidence that cis-trans isomerization is not occurring in the solid state (i.e. in the cuticle) but that changes in molecular geometry are still occurring that could enable energy dissipation. However, I see rather little evidence for the relative contributions of these two mechanisms to the overall dissipation of the absorbed UV energy, just the statement about quantum yield (lines 158-159). The authors state in several places that the non-radiative mechanism predominates. What is the rationale for those statements? The manuscript should be modified to emphasize and make that rationale very clear.

Quantum yield of *p*-coumaric acid ranges between 0.00001 and 0.0001 (ref 32 in the ms), whereas other plant fluorescent molecules such as chlorophyll a and b have values around 0.25 and 0.11 respectively (Forster and Livingston Journal of Chemical Physics 20 (1952) 1315-1320). If we combine this low quantum yield (1 molecule out of 100000 or 10000 will dissipate energy emitting fluorescence) with the low fluorescence detected in cross-sections and in fluorescence spectra of cuticles and their lifetimes (see Suppl Fig 3 and 4 and lines 157-160), it is expected that another mechanism or mechanisms (vibrational relaxation, intersystem crossing etc) are responsible for most of the energy dissipation. This is the rationale we mentioned in lines 160-164 in the ms.

However, it is true that in the legend of Fig 5 it was indicated that trans-cis isomerization was the main non-radiative event and we have corrected that (see the ms) since others such as vibrational relaxation, intersystem crossing etc. can also be participating, as we indicated in lines 375-378.

Minor comments

Lines 140: Since none of the aliphatic and terpenoid compounds of waxes have conjugated double bonds. This is true for some species, but not all. Alkyl resorcinols, found in a variety of plant species wax mixtures, contain conjugated double bonds. Please modify the sentence to make it more accurate.

It was our intention to indicate that long chain alcohol, alkane or terpenoid do not contain conjugated double bonds by itself. Unless it is chemically bonded to a phenolic compound, of course. But the referee has raised a valid point. We have removed the sentence to avoid confusion since we realized it can be more confusing than helpful.

The article needs to be thoroughly proofread. Many errors, including some that should have been caught by an automated spell-checking tool, can be found throughout the manuscript. I list a few here, but the list is non-exhaustive:

Line 85: present in the cuticle have already shown to play..... present in the cuticle have already been shown to play.....

Line 176: p-coumaric was also analysed; p-coumaric acid was also analysed

Line 224: corresponded to the cis-isomer However,...corresponded to the cis-isomer. However; (missing period)

The ms has been carefully checked and the misspellings corrected, including those indicated by the referee.

Phenolic acids present in the cell walls, which the authors suggest may absorb UV light, are not present in Fig. 5. The authors should consider adding them to make the figure more comprehensive.

The idea suggested is quite interesting. However, we are only indicating the possibility of cell walls carrying an additional protective layer. But we feel that this topic would need to be properly investigated before including it in Figure 5, otherwise it could be misinterpreted.

Line 70/71: Waxes can have a dual location, either vanishing the surface (epicuticular) what does vanishing mean here? Do the authors mean that the waxes vanish to the surface? If so, I am not sure vanishing is the right word, since waxes are particularly visible when on the surface. Perhaps change this word to help improve reader comprehension.

The word varnish was clearly misspelled. We have however modified the sentence since the verb could be removed and thus avoid any confusion. See line 71.

Line 51: cope : overcome.

It has been modified. See line 51.

Quite a few places are missing citations. Please search the manuscript for all such instances and correct this. Some starting points are: (i) lines 76-78, ending in acting a thermal regulator, (ii) lines 148-150, Of the different phenolics present in plant cuticles, p-coumaric acid has been identified ubiquitously and in some instances as the only cinnamic acid derivative and (iii) lines 126-128, Epicuticular wax disposition and crystallization, as well as the presence and density of indument, are chiefly responsible for light reflection and could be behind the observed differences among species.

We have included the corresponding references and checked the ms to provide adequate missing references.

Line 112: reduced cuticles. Do the authors mean reduced abundances of phenolics? Reduced cuticles could refer to wax, cutin, phenolics, or all the above. Please be more specific.

It was referring to the amount of cuticle, which has now been clearly indicated in the sentence (see lines 111-112).

While reading the first section of the results section, the first thing I thought of was what does the absorption spectrum of a p-coumaric acid standard look like?. Would it make more sense to introduce the absorption characteristics of the p-coumaric acid standard at the beginning of the first results section, rather than at the end of that section? Just an idea.

The referee is right and we have moved this description of the p-coumaric acid absorption spectra to be introduced right after describing cuticle absorption. This way the comparison is much clearer (see lines 139-146).

Line 375-378 This ultrafast dissipative mechanism of UV energy, in combination with the much slower fluorescence emission and other non-radiative pathways such as vibrational relaxation act in combination to protect plant tissues from

harmful UV radiation. This is a rather strong statement that could, in the future, be rendered incomplete or even inaccurate, for example, if we were to learn more about the absorption properties of phenolics inside the cuticle other than p-coumaric acid. Perhaps seem to act in combination would be a safer and more appropriate configuration for this sentence.

The referee is right. The identification of cuticles with phenolic compounds, other than cinnamates, may show modified absorption properties. The sentence has been modified accordingly (see line 377).

Line 401: basal plants. In this context, what does basal mean? Does it mean that the branches of the plant phylogeny leading to bryophytes are at the bottom of the tree? If this is what is meant, then that is inaccurate (see: McDaniel, S.F., 2021. Bryophytes are not early diverging land plants. New Phytologist, 230(4), pp.1300-1304.). The argument the authors appear to be making is that phenolics are found in diverse land plants. If this is the case, then I think substituting the word basal for the word diverse will both strengthen the authors argument and make the sentence more accurate.

The referee is right. We have modified the sentence to simply indicate the plant taxonomic groups to which the species studied in the cited papers belong. This would avoid any potential confusion with the second half of the sentence indicating a notable abundance of phenolics in those species (see line 401).

REVIEWER COMMENTS

Reviewer #1 (Remarks to the Author):

The authors have made a good effort to address my comments, along with the other reviewers, I still have some small concerns regarding the quality of the transient absorption data analysis. For instance the fit for coumaric acid in methanol only shows two EADS (The omission of t_1 is understandable given the IRF of the system), this suggests only three time-constants were used, while the residual clearly shows that a fourth would be required to model an absorption feature at 360ish nm which is the cis-isomer being formed (as the authors have ruled out the phenolic radical via power dependence studies). Furthermore in some cases the authors have quoted errors on the time-constants that are less than their instrument response. While the fit might provide a lower error, ultimately the IRF determines the minimum certainty you can have on a time-constant ($\text{IRF}/2$).

Ultimately, these are easy fixes and do not change the story of the paper, I feel they are necessary before the paper should be accepted for publication.

Reviewer #2 (Remarks to the Author):

The authors have addressed extensively the comments by the reviewers. Nevertheless, for me an important argument remains that from a spectroscopic point of view interesting results are reported but nothing with a major impact. The overruling reason for publication in Nature Communications would thus need to be that from a biological point of view the present results would lead to completely new insights. The authors have not been able to convince me of such a conclusion.

Apart from this general observation, there also still remain quite a number of other issues.

1. In their response to Reviewer 1 the authors mention that they have performed a sequential fit of the TAS, and that this has led to different lifetimes (although remaining in the same range as with a global analysis). This cannot be the case: Glotaran fits the data with a sum of decaying exponentials, and this fit is independent of the target model that is subsequently used.

2. The authors mention that that for coumaric acid in methanol stimulated emission is observed. My problem with Fig. 2 (and Fig. S6) is that there appears to be a very broad stimulated emission continuum from 450-700 nm at short delay times that is quite different from the much more narrow feature at longer times. Moreover, in the 650-700 nm region there appears still to be an induced absorption feature.

3. I am a bit skeptical about Figs. S13 and 14. First of all, it would be good to put in error bars, but apart from that I think that four (very scattered) points is too weak a basis to conclude whether absorption is linear in intensity or not. In fact, in Fig. S14 slopes of 1.3-1.4 are found which should set off alarm bells.
4. Figs. S6, 7, and 9 have been adapted to show residual heat maps. If one looks at the scale of these residual heat maps, these appear to go in certain regions to -6 to -9 mOD, implying deviations of 20-25% from the measured values, which are much larger than one normally sees. In order to obtain a good impression of how well the data are fitted it would be very useful to show in Fig.2 and the Supplementary Figures the experimental and fitted time traces and not only the experimental ones. One further remark on Fig. 2: the color scale for the methanol solution heat map (ranging between -2 and 4) does not match the y-axis in the time traces below the heat map.
5. The authors report that they find fluorescence lifetimes for the cuticles between 2 and 8 ns. This immediately shows that the emission cannot originate from the same species measured in the TAS because from the TAS it is concluded that internal conversion to the ground state (τ_3) occurs on a time scale of a few 100's of picoseconds. Fig. S4 also shows that the emission spectra are vastly different from one type of cuticle to another.
6. τ_2 is -apart from pCA solution- for all samples attributed to vibrational cooling and overcoming the sp-TS-S1 barrier, while τ_3 is associated with internal conversion to the ground state. Table 1 is intriguing in that pCA oligomer and *Beta vulgaris* show significantly larger τ_2 values. Is it possible to find an explanation for that?
7. The authors indicate that they included dynamic electron correlation by performing MP2 calculations. Such calculations only take dynamic electron correlation in the ground state into account, where it is not expected to be so important. On the contrary, for the electronically excited states dynamic electron correlation is expected to play a much larger role, but for these states one really needs CASPT2 calculations.
8. Figure S2 shows a curvilinear regression fit of the absorbance versus the amount of cuticle phenolics. There should be a linear relation between the two according to Beer's law, so what is the reason that this is not the case?

Reviewer #3 (Remarks to the Author):

The authors have addressed all of my comments satisfactorily. I congratulate them on their hard work and suggest that the manuscript be published.

Reviewer#1

The authors have made a good effort to address my comments, along with the other reviewers, I still have some small concerns regarding the quality of the transient absorption data analysis. For instance the fit for coumaric acid in methanol only shows two EADS (The omission of t_1 is understandable given the IRF of the system), this suggest only three time-constants were used, while the residual clearly shows that a fourth would be required to model an absorption feature at 360ish nm which is the cis-isomer being formed (as the authors have ruled out the phenolic radical via power dependence studies).

The referee is right, τ_4 ascribed to the presence of the cis-isomer was missed in the fitting of the solution of *p*-coumaric acid. We have added this new component and the ms has been modified at

Furthermore in some cases the authors have quoted errors on the time-constants that are less than their instrument response. While the fit might provide a lower error, ultimately the IRF determines the minimum certainty you can have on a time-constant (IRF/2). Ultimately, these are easy fixes and do not change the story of the paper, I feel they are necessary before the paper should be accepted for publication.

Thanks for your comments along this evaluation process. The errors on time-constants that were smaller than what could be accurately calculated limited by the IRF have been modified in Table 1.

Reviewer #2

The authors have addressed extensively the comments by the reviewers. Nevertheless, for me an important argument remains that from a spectroscopic point of view interesting results are reported but nothing with a major impact. The overruling reason for publication in Nature Communications would thus need to be that from a biological point of view the present results would lead to completely new insights. The authors have not been able to convince me of such a conclusion.

As we indicated in our last version of the ms there is good research on photoprotection of cinnamic acids derivatives but most of it has been made on these molecules in solution. Our study has been focused on isolated cuticles (solid biological samples) containing one of these cinnamic acids (*p*-coumaric) at low amounts and with a potential critical role in plant photoprotection against UV radiation that has never been investigated in detail. Most of the research on this subject in the plant physiology and biophysics fields take into account the presence of phenolics within cells but never at the outermost cuticle level. Our ms demonstrated that isolated cuticles from leaves and fruits of a broad range of plant species (gymnosperm, angiosperm, monocots, dicots, ferns) show a high degree of photoprotection against UV radiation following a similar photophysical mechanism. These conclusions go beyond a spectroscopic study and has been possible after the interdisciplinary work of researchers from botany, quantum chemistry and spectroscopy fields. Thus, our study can be considered a significant advance in the understanding of the role of phenolics in the biophysics of plant cuticles (<https://doi.org/10.1111/j.1469-8137.2010.03553.x>).

Moreover, in the current scenario of climate change, plant breeding programs have to consider increases in irradiation (especially UV light) that could be harmful for crop production and develop strategies to mitigate their impact. In this sense, identification of genes involved in phenolic accumulation are gaining a lot of interest as an approach to increase the amount of phenolics within cells and protect plants against harmful UV radiation. However, they are failing to consider the cuticle as the first barrier against UV light, a barrier that reduces over 50% of UV-B light! (and in some cases almost completely block UV-B light transmittance). This knowledge provides new insights into plant protection from harmful radiation and on the development of novel strategies for their improvement. For this purpose, comprehending the photochemical and photophysical mechanisms of phenolics in the cuticles will be essential.

Finally, we would like to indicate that the true value of a scientific work lies more in what it suggests and how it helps formulating future questions. In this sense we really believe that the present ms open new perspectives to investigate the

still unknown role of phenolics in other supramolecular plant structures as cell walls, lignin, sporopollenin or suberin.

Apart from this general observation, there also still remain quite a number of other issues.

1. In their response to Reviewer 1 the authors mention that they have performed a sequential fit of the TAS, and that this has led to different lifetimes (although remaining in the same range as with a global analysis). This cannot be the case: Glotaran fits the data with a sum of decaying exponentials, and this fit is independent of the target model that is subsequently used.

The changes in the calculated lifetimes are not due to the change from a parallel to the sequential model itself, but from the change in the software employed. In the first version of the ms, we used the software *Surface Explorer* to calculate lifetimes of the photodynamic processes. Nevertheless, referee 1 requested the use of a sequential model. Since *Surface Explorer* only offers the possibility to carry out global fitting analyses with a parallel model, we changed to the software Glotaran to recalculate the dynamics considering a sequential model, as we mentioned in our response to referee 1. The differences of lifetimes observed between the parallel and sequential models are associated to this change of software. While *Surface Explorer* use the nonlinear Marquardt algorithm to fit multi-exponential decay law convoluted with the Gaussian instrument response function, Glotaran uses the variable projection algorithm. We apologize if the referee was misled by our response to referee 1.

2. The authors mention that that for coumaric acid in methanol stimulated emission is observed. My problem with Fig. 2 (and Fig. S6) is that there appears to be a very broad stimulated emission continuum from 450-700 nm at short delay times that is quite different from the much more narrow feature at longer times. Moreover, in the 650-700 nm region there appears still to be an induced absorption feature.

We are very thankful to the referee for spotting this error in Figure 2. There was a problem during data export of the black spectrum, which corresponds to 197 fs delay time, and it had an incorrect offset. As it can be observed in the modified Figure 2, the black spectrum does not show any stimulated emission. The only spectra which shows SE (around 400 to 500 nm) is the red one (at 766 fs).

3. I am a bit skeptical about Figs. S13 and 14. First of all, it would be good to put in error bars, but apart from that I think that four (very scattered) points is too

weak a basis to conclude whether absorption is linear in intensity or not. In fact, in Fig. S14 slopes of 1.3-1.4 are found which should set off alarm bells.

New power dependence studies have been carried out with 6 different powers and error bars have been included (see new Figs. S16 and 17). Results indicate one-photon dynamics in all the instances.

4. Figs. S6, 7, and 9 have been adapted to show residual heat maps. If one looks at the scale of these residual heat maps, these appear to go in certain regions to -6 to -9 mOD, implying deviations of 20-25% from the measured values, which are much larger than one normally sees. In order to obtain a good impression of how well the data are fitted it would be very useful to show in Fig.2 and the Supplementary Figures the experimental and fitted time traces and not only the experimental ones. One further remark on Fig. 2: the color scale for the methanol solution heat map (ranging between -2 and 4) does not match the y-axis in the time traces below the heat map.

In the previous version of the ms, the scales employed in the residual maps were chosen to display the same colour for the zero value in all of them, thus allowing for a much easier comparison among the different heat maps. To achieve this purpose, additional upper and/or lower values were added to the scale in some maps, as the graph software does not allow to choose individual colours but colour ranges. Thus, some cases, the upper and/or lower scale colours did not represent any value present in the map. For us, this was clearly obvious since no purple or red were present in some maps. However, we understand that small changes in colour, especially in certain ranges of the spectrum, can be tricky to spot, even more so after resizing the maps to fit several within a Figure. For this reason, we have decided to use scales within the range value for each residual map (see new Figs. S6, S7 and S9). Now, each residual map displays all the colours present in its own scale. However, due to the difficulties associated with colour vision and object size, it is worth mentioning that the highest values of the residual scales are only observed as discrete points in the maps, usually at the lowest wavelengths close to the excitation wavelength of the pump beam or as individual points in solid samples due to scatter light noise. Additionally, following the referee's suggestion, we have included the experimental and fitted time traces (see new Figs.S10, S11 and S12) to show the goodness of the fittings.

The referee is right, there was an error in the colour scale of the heat map of the *p*-coumaric acid solution which has been properly modified (see Fig.2).

5. The authors report that they find fluorescence lifetimes for the cuticles between 2 and 8 ns. This immediately shows that the emission cannot originate from the same species measured in the TAS because from the TAS it is concluded that

internal conversion to the ground state (tau3) occurs on a time scale of a few 100's of picoseconds. Fig. S4 also shows that the emission spectra are vastly different from one type of cuticle to another.

Spectra assigned to tau2 and tau3 are ascribed to excited-state dynamic processes and not to fluorescence emission, as it was already indicated in the ms. In typical TAS experiment only a very low fraction of the molecules are promoted to an electronically excited state. Also, only the radiation in the pump-probe pathway can be recorded. However, fluorescence is a spontaneous emission where the phase of the photon is random, as it is the direction of photon propagation. All the above, combined with a low quantum yield, minimizes the probability of registering this process in TAS. Additionally, the broad TAS band observed in solid samples could be overlapping with the spontaneous/stimulated emissions and nullifying them.

Truly, emission spectra of some cuticles differ in fluorescence intensity, usually low, and shape. This is the result of different amounts of *p*-coumaric acid and variability in its location and arrangement into waxes and/or cutin matrix of the corresponding cuticle. Molecular environments of different polarity, structural and lamellar rigidity have significant influence on the fluorescence characteristics. The referee should also keep in mind that we are comparing cuticles from different organs (leaves and fruits) belonging to different species and plant families, to six different plant orders and two different plant classes; therefore, variability is not surprising.

6. Tau2 is -apart from pCA solution- for all samples attributed to vibrational cooling and overcoming the sp-TS-S1 barrier, while tau3 is associated with internal conversion to the ground state. Table 1 is intriguing in that pCA oligomer and Beta vulgaris show significantly larger tau2 values. Is it possible to find an explanation for that?

As it was mentioned in the manuscript, these times are an indication of phenolics present in a more rigid and restricted structural environment (see lines 212-217 and 245-253). Reported structure and physical properties on the oligomer indicate a rigid matrix with presence of aliphatic and aromatics domains (<https://doi.org/10.1371/journal.pone.0214956>). In the case of *B. vulgaris* it was mentioned that the cuticle is mainly composed of cutan instead of cutin. Cutan is a more rigid polymer, derived from unsaturated fatty acids and cross-linked with ether and peroxide bonds. This polymer rigidity would hinder molecular movements within the matrix and could explain the notably high lifetimes detected in the *B. vulgaris* (lines 245-253). A small modification to the text was added to remind the similarly large lifetimes observed in the oligomer and *B. vulgaris* cuticle (see lines 252-253)

7. The authors indicate that they included dynamic electron correlation by performing MP2 calculations. Such calculations only take dynamic electron correlation in the ground state into account, where it is not expected to be so important. On the contrary, for the electronically excited states dynamic electron correlation is expected to play a much larger role, but for these states one really needs CASPT2 calculations.

According to the reviewer comments, we have included CASPT2/cc-TZV single point energy calculation in order to consider dynamic correlation effects in the revised version of the manuscript. For that, we used triple zeta Dunning's basis set because 6-31+G(d,p) is not implemented in the OpenMolcas program. Our results show that dynamic electron correlation computed by CASPT2 single point energy has a significant effect on the vertical excitations energy and the S1 potential energy surface in solution. However, the general results are almost identical to the ones obtained when MP2 corrections were considered, but the computational time spent was c.a. 10 times longer. The conical intersections obtained within CASSCF(11,12) or CASPT2 energetic spaces are geometrically similar in both scenarios, therefore no modifications on the general conclusions were necessary. The main manuscript (lines 307-314) and the supporting information (Supplementary Table 1 and 2) have been modified in order to include these news calculations.

8. Figure S2 shows a curvilinear regression fit of the absorbance versus the amount of cuticle phenolics. There should be a linear relation between the two according to Beer's law, so what is the reason that this is not the case?

Figure S2 was included as a request of referee # 3 to display a regression between cuticle phenolic abundance in different species and their cuticle absorbance within the UV-B region. The plot showed a significant relationship between the two variables. In any case, this cannot be interpreted as a Lambert-Beer representation or as the result of this law. For this to be the case, cuticle thickness (extraordinary variable among the species studied) and phenolic arrangement (free of forming clusters) within the cuticle would have to be considered for each species.

Reviewer#3

The authors have addressed all of my comments satisfactorily. I congratulate them on their hard work and suggest that the manuscript be published.

The authors thank the referee's comments which have contributed to improve the ms.

REVIEWERS' COMMENTS

Reviewer #1 (Remarks to the Author):

The Authors have addressed all my comments

Reviewer #2 (Remarks to the Author):

The authors have put significant effort to address the comments. As I said initially, I am not a biologist and am only able to judge the spectroscopic part of the work. The authors make a case with their comments in the present rebuttal with which I am willing to go along. The authors have addressed the spectroscopic comments convincingly. The only remark I still would like to make concerns the fluorescence lifetimes. The authors assign τ_3 as associated with internal conversion to the ground state. Since this implies that the lifetime of the electronically excited state is thus (significantly) less than 1 ns, it cannot have a fluorescence lifetime between 2 and 8 ns. These observations suggest that branching occurs on the potential energy surface of the electronically excited state.

Reviewer#1

The authors thank the referee's comments which have contributed to improve our ms.

Reviewer#2

The authors have put significant effort to address the comments. As I said initially, I am not a biologist and am only able to judge the spectroscopic part of the work. The authors make a case with their comments in the present rebuttal with which I am willing to go along. The authors have addressed the spectroscopic comments convincingly. The only remark I still would like to make concerns the fluorescence lifetimes. The authors assign tau3 as associated with internal conversion to the ground state. Since this implies that the lifetime of the electronically excited state is thus (significantly) less than 1 ns, it cannot have a fluorescence lifetime between 2 and 8 ns. These observations suggest that branching occurs on the potential energy surface of the electronically excited state.

The authors thank the referee's comments along this evaluation process, they have added notable value to our ms. As the Reviewer 2 points, tau3 is significantly less than 1 ns (data shown in Table 1) for all the studied species. Nevertheless, Table 1 also includes a tau4 which is longer than 2 ns for all the samples. We are not able to estimate the exact lifetime value of this tau4 due to the time scale limitation in TAS measurements. This long lifetime indicates the existence of a population of molecules which remain excited some nanoseconds after the excitation pulse. We agree with the referee in a plausible branching on the potential energy surface of the electronically excited state allowing the deactivation of the molecules by both, radiative and non-radiative pathway. The coexistence of both mechanism is reflected in Figure 5 of the ms.